# Wasserstein $K$-means for clustering probability distributions

**Yubo Zhuang, Xiaohui Chen, Yun Yang**
Department of Statistics
University of Illinois at Urbana-Champaign
{yubo2,xhchen,yy84}@illinois.edu

## Abstract

Clustering is an important exploratory data analysis technique to group objects based on their similarity. The widely used $K$-means clustering method relies on some notion of distance to partition data into a fewer number of groups. In the Euclidean space, centroid-based and distance-based formulations of the $K$-means are equivalent. In modern machine learning applications, data often arise as probability distributions and a natural generalization to handle measure-valued data is to use the optimal transport metric. Due to non-negative Alexandrov curvature of the Wasserstein space, barycenters suffer from regularity and non-robustness issues. The peculiar behaviors of Wasserstein barycenters may make the centroid-based formulation fail to represent the within-cluster data points, while the more direct distance-based $K$-means approach and its semidefinite program (SDP) relaxation are capable of recovering the true cluster labels. In the special case of clustering Gaussian distributions, we show that the SDP relaxed Wasserstein $K$-means can achieve exact recovery given the clusters are well-separated under the 2-Wasserstein metric. Our simulation and real data examples also demonstrate that distance-based $K$-means can achieve better classification performance over the standard centroid-based $K$-means for clustering probability distributions and images.

## 1 Introduction

Clustering is a major tool for unsupervised machine learning problems and exploratory data analysis in statistics. Suppose we observe a sample of data points $X_1, \ldots, X_n$ taking values in a metric space $(\mathcal{X}, \|\cdot\|)$. Suppose there exists a clustering structure $G_1^*, \ldots, G_K^*$ such that each data point $X_i$ belongs to exactly one of the unknown cluster $G_k^*$. The goal of clustering analysis is to recover the true clusters $G_1^*, \ldots, G_K^*$ given the input data $X_1, \ldots, X_n$. In the Euclidean space $\mathcal{X} = \mathbb{R}^p$, the $K$-means clustering is a widely used method that achieves the empirical success in many applications [MacQueen, 1967]. In modern machine learning and data science problems such as computer graphics [Solomon et al., 2015], data exhibits complex geometric features and traditional clustering methods developed for Euclidean data may not be well suited to analyze such data.

In this paper, we consider the clustering problem of probability measures $\mu_1, \ldots, \mu_n$ into $K$ groups. As a motivating example, the MNIST dataset contains images of handwritten digits 0-9. Normalizing the greyscale images into histograms as probability measures, a common task is to cluster the images. One can certainly apply the Euclidean $K$-means to the vectorized images. However, this would lose important geometric information of the two-dimensional data. On the other hand, theory of optimal transport [Villani, 2003] provides an appealing framework to model measure-valued data as probabilities in many statistical tasks [Domazakis et al., 2019, Chen et al., 2021, Bigot et al., 2017, Seguy and Cuturi, 2015, Rigollet and Weed, 2019, Hütter and Rigollet, 2019, Cazelles et al., 2018].

36th Conference on Neural Information Processing Systems (NeurIPS 2022).

**Background on $K$-means clustering.** Algorithmically, the $K$-means clustering have two equivalent formulations in the Euclidean space – centroid-based and distance-based – in the sense that they both yield the same partition estimate for the true clustering structure. Given the Euclidean data $X_1, \ldots, X_n \in \mathbb{R}^p$, the *centroid-based* formulation of standard $K$-means can be expressed as

$$\min_{\beta_1,\ldots,\beta_K \in \mathbb{R}^d} \sum_{i=1}^n \min_{k \in [K]} \|X_i - \beta_k\|_2^2 = \min_{G_1,\ldots,G_K} \left\{ \sum_{k=1}^K \sum_{i \in G_k} \|X_i - \bar{X}_k\|_2^2 : \bigsqcup_{k=1}^K G_k = [n] \right\}, \quad (1)$$

where clusters $\{G_k\}_{k=1}^K$ are determined by the Voronoi diagram from $\{\beta_k\}_{k=1}^K$, $\bar{X}_k = |G_k|^{-1} \sum_{i \in G_k} X_i$ denotes the centroid of cluster $G_k$, $\bigsqcup$ denotes the disjoint union and $[n] = \{1, \ldots, n\}$. Heuristic algorithm for solving (1) includes Lloyd's algorithm [Lloyd, 1982], which is an iterative procedure alternating the partition and centroid estimation steps. Specifically, given an initial centroid estimate $\beta_1^{(1)}, \ldots, \beta_K^{(1)}$, one first assigns each data point to its nearest centroid at the $t$-th iteration according to the Voronoi diagram, i.e.,

$$G_k^{(t)} = \left\{ i \in [n] : \|X_i - \beta_k^{(t)}\|_2 \leqslant \|X_i - \beta_j^{(t)}\|_2, \ \forall j \in [K] \right\}, \quad (2)$$

and then update the centroid for each cluster

$$\beta_k^{(t+1)} = \frac{1}{|G_k^{(t)}|} \sum_{i \in G_k^{(t)}} X_i, \quad (3)$$

where $|G_k^{(t)}|$ denotes the cardinality of $G_k^{(t)}$. Step (2) and step (3) alternate until convergence.

The *distance-based* (sometimes also referred as *partition-based*) formulation directly solves the following constrained optimization problem without referring to the estimated centroids:

$$\min_{G_1,\ldots,G_K} \left\{ \sum_{k=1}^K \frac{1}{|G_k|} \sum_{i,j \in G_k} \|X_i - X_j\|_2^2 : \bigsqcup_{k=1}^K G_k = [n] \right\}. \quad (4)$$

Observe that (1) with nearest centroid assignment and (4) are equivalent for the clustering purpose due to the following identity, which extends the parallelogram law from two points to $n$ points,

$$\sum_{i,j=1}^n \|X_i - X_j\|_2^2 = 2n \sum_{i=1}^n \|X_i - \bar{X}\|_2^2, \quad \text{with} \quad \bar{X} = \frac{1}{n} \sum_{i=1}^n X_i \quad \text{and} \quad X_i \in \mathbb{R}^p. \quad (5)$$

Consequently, the two criteria yield the same partition estimate for $G_1^*, \ldots, G_K^*$. The key identity (5) establishing the equivalence relies on two facts of the Euclidean space: (i) it is a vector space (i.e., vectors can be averaged in the linear sense); (ii) it is flat (i.e., zero curvature), both of which are unfortunately not true for the Wasserstein space $(\mathcal{P}_2(\mathbb{R}^p), W_2)$ that endows the space $\mathcal{P}_2(\mathbb{R}^p)$ of all probability distributions with finite second moments with the 2-Wasserstein metric $W_2$ [Ambrosio et al., 2005]. In particular, the 2-Wasserstein distance between two distributions $\mu$ and $\nu$ in $\mathcal{P}_2(\mathbb{R}^p)$ is defined as

$$W_2^2(\mu, \nu) := \min_{\gamma} \left\{ \int_{\mathbb{R}^p \times \mathbb{R}^p} \|x - y\|_2^2 \, \mathrm{d}\gamma(x, y) \right\}, \quad (6)$$

where minimization over $\gamma$ runs over all possible couplings with marginals $\mu$ and $\nu$. It is well-known that the Wasserstein space is a metric space (in fact a geodesic space) with non-negative curvature in the Alexandrov sense [Lott, 2008].

**Our contributions.** We summarize our main contributions as followings: (i) we provide evidence for pitfalls (irregularity and non-robustness) of barycenter-based Wasserstein $K$-means, both theoretically and empirically, and (ii) we generalize the distance-based formulation of $K$-means to the Wasserstein space and establish the exact recovery property of its SDP relaxation for clustering Gaussian measures under a separateness lower bound in the 2-Wasserstein distance.

**Existing work.** Since the $K$-means clustering is a worst-case NP-hard problem [Aloise et al., 2009], approximation algorithms have been extensively studied in literature including: Lloyd's algorithm [Lloyd, 1982], spectral methods [von Luxburg, 2007, Meila and Shi, 2001, Ng et al., 2001], semidefinite programming (SDP) relaxations [Peng and Wei, 2007], non-convex methods via

low-rank matrix factorization [Burer and Monteiro, 2003]. Theoretic guarantees of those methods are established for statistical models on Euclidean data [Lu and Zhou, 2016, von Luxburg et al., 2008, Vempala and Wang, 2004, Fei and Chen, 2018, Giraud and Verzelen, 2018, Chen and Yang, 2021, Zhuang et al., 2022].

The concept of clustering general measure-valued data is introduced by Domazakis et al. [2019], where the authors proposed the centroid-based Wasserstein K-means algorithm. It replaced the Euclidean norm and sample means by the Wasserstein distance and barycenters respectively. Verdinelli and Wasserman [2019] proposed a modified Wasserstein distance for distribution clustering. And after that, Chazal et al. [2021] proposed a method in Clustering of measures via mean measure quantization by first vectorizing the measures in a finite Euclidean space followed by an efficient clustering algorithm such as single-linkage clustering with $L_\infty$ distance. The vectorization methods could improve the computational efficiency but might not capture the properties of probability measures well compared to clustering algorithms based directly on Wasserstein space.

## 2 Wasserstein $K$-means clustering methods

In this section, we generalize the Euclidean $K$-means to the Wasserstein space. Our starting point is to mimic the standard $K$-means methods for Euclidean data. Thus we may define two versions of the Wasserstein $K$-means clustering formulations: *centroid-based* and *distance-based*. As we mentioned in Section 1, when working with Wasserstein space $(\mathcal{P}_2(\mathbb{R}^p), W_2)$, the corresponding centroid-based criterion (1) and the distance-based criterion (4), where the Euclidean metric $\|\cdot\|_2$ is replaced with the 2-Wasserstein metric $W_2$, may lead to radically different clustering schemes. To begin with, we would like to argue that due to the irregularity and non-robustness of barycenters in the Wasserstein space, the centroid-based criterion may lead to unreasonable clustering schemes that lack physical interpretations and are sensitive to small data perturbations.

### 2.1 Clustering based on barycenters

The centroid-based Wasserstein $K$-means for extending the Lloyd's algorithm into the Wasserstein space has been recently considered by Domazakis et al. [2019]. Specifically, it is an iterative algorithm proceeds as following. Given an initial centroid estimate $\nu_1^{(1)}, \ldots, \nu_K^{(1)}$, one first assigns each probability measure $\mu_1, \ldots, \mu_n$ to its nearest centroid in the Wasserstein geometry at the $t$-th iteration according to the Voronoi diagram:

$$G_k^{(t)} = \left\{ i \in [n] : W_2(\mu_i, \nu_k^{(t)}) \leqslant W_2(\mu_i, \nu_j^{(t)}), \quad \forall j \in [K] \right\}, \tag{7}$$

and then update the centroid for each cluster

$$\nu_k^{(t+1)} = \arg\min_{\nu \in \mathcal{P}_2(\mathbb{R}^d)} \frac{1}{|G_k^{(t)}|} \sum_{i \in G_k^{(t)}} W_2^2(\mu_i, \nu). \tag{8}$$

Note that $\nu_k^{(t+1)}$ in (8) is referred as *barycenter* of probability measures $\mu_i, i \in G_k^{(t)}$, a Wasserstein analog of the Euclidean average or mean [Agueh and Carlier, 2011]. We will also ex-changeably use barycenter-based $K$-means to mean the centroid-based K-means in the Wasserstein space. Even though the Wasserstein barycenter is a natural notion of averaging probability measures, it may exhibit peculiar behaviours and fail to represent the within-cluster data points, partly due to the violation of the generalized parallelogram law (5) (for non-flat spaces) if the Euclidean metric $\|\cdot\|_2$ is replaced with the 2-Wasserstein metric $W_2$.

*Example* 1 (**Irregularity of Wasserstein barycenters**). Wasserstein barycenter has much less regularity than the sample mean in the Euclidean space [Kim and Pass, 2017]. In particular, Santambrogio and Wang [2016] constructed a simple example of two probability measures that are supported on line segments in $\mathbb{R}^2$, whereas the support of their barycenter obtained as the displacement interpolation the two endpoint probability measures is not convex (cf. left plot in Figure 1). In this example, the probability density $\mu_0$ and $\mu_1$ are supported on the line segments $L_0 = \{(s, as) : s \in [-1, 1]\}$ and $L_1 = \{(s, -as) : s \in [-1, 1]\}$ respectively. We choose $a \in (0, 1)$ to identify the orientation of $L_0$ and $L_1$ based on the $x$-axis. Moreover, we consider the linear density functions $\mu_0(s) = (1-s)/2$

and $\mu_1(s) = (1+s)/2$ for $s \in [-1,1]$ supported on $L_0$ and $L_1$ respectively. Then the optimal transport map $T := T_{\mu_0 \to \mu_1}$ from $\mu_0$ to $\mu_1$ is given by

$$T(x, ax) = \left(-1 + \sqrt{4 - (1-x)^2}, \ -a \cdot (-1 + \sqrt{4 - (1-x)^2})\right), \tag{9}$$

and the barycenter corresponds to the displacement interpolation $\mu_t = [(1-t)\mathrm{id} + tT]_\sharp \mu_0$ at $t = 0.5$ [McCann, 1997]. For self-contained purpose, we give the proof of (9) in Appendix D.1. Fig. 1 on the left shows the support of barycenter $\mu_{0.5}$ is not convex (in fact part of an ellipse boundary) even though the supports of $\mu_0$ and $\mu_1$ are convex. This example shows that the barycenter functional is not geodesically convex in the Wasserstein space. As barycenters turn out to be essential in centroid-based Wasserstein $K$-means and irregularity of the barycenter may fail to represent the cluster (see more details in Example 3 and Remark 9 below), this counter-example is our motivation to seek alternative formulation. ∎

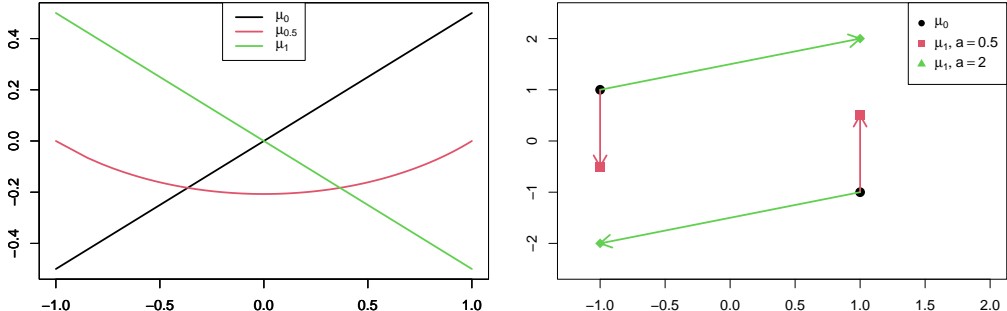

Figure 1: Left: support of the Wasserstein barycenter as the displacement interpolation between $\mu_0$ and $\mu_1$ at $t = 0.5$ in Example 1. Right: non-robustness of the optimal transport map (arrow lines) and Wasserstein barycenter w.r.t. small perturbation around $a = 1$ for the target measure in Example 2.

*Example* 2 (**Non-robustness of Wasserstein barycenters**). Another unappealing feature of the Wasserstein barycenter is its sensitivity to data perturbation: a small (local) change in one contributing probability measure may lead to large (global) changes in the resulting barycenter. See Fig. 1 on the right for such an example. In this example, we take the source measure as $\mu_0 = 0.5\,\delta_{(-1,1)} + 0.5\,\delta_{(1,-1)}$ and the target measure as $\mu_1 = 0.5\,\delta_{(-1,-a)} + 0.5\,\delta_{(1,a)}$ for some $a > 0$. It is easy to see that the optimal transport map $T := T_{\mu_0 \to \mu_1}$ has a dichotomy behavior:

$$T(-1, 1) = \begin{cases} (-1, -a) & \text{if } 0 < a < 1 \\ (1, a) & \text{if } a > 1 \end{cases} \quad \text{and} \quad T(1, -1) = \begin{cases} (1, a) & \text{if } 0 < a < 1 \\ (-1, -a) & \text{if } a > 1 \end{cases}. \tag{10}$$

Thus the Wasserstein barycenter determined by the displacement interpolation $\mu_t = [(1-t)\mathrm{id} + tT]_\sharp \mu_0$ is a discontinuous function at $a = 1$. This non-robustness can be attribute to the discontinuity of the Wasserstein barycenter as a function of its input probability measures; in contrast, the Euclidean mean is a globally Lipchitz continuous function of its input points. ∎

Because of these pitfalls of the Wasserstein barycenter shown in Examples 1 and 2, the centroid-based Wasserstein $K$-means approach described at the beginning of this subsection may lead to unreasonable and unstable clustering schemes. In addition, an ill-conditioned configuration may significantly slow down the convergence of commonly used barycenter approximating algorithms such as iterative Bregman projections [Benamou et al., 2015]. Below, we give a concrete example of such phenomenon in the clustering context.

*Example* 3 (**Failure of centroid-based Wasserstein $K$-means**). In a nutshell, the failure in this example is due to the counter-intuitive phenomenon illustrated in the right panel of Fig. 2, where some distribution $\mu_3$ in the Wasserstein space may have larger $W_2$ distance to Wasserstein barycenter $\mu_1^*$ than every distribution $\mu_i$ ($i = 1, 2$) that together forms it. As a result of this strange configuration, even though $\mu_3$ is closer to $\mu_1$ and $\mu_2$ from the first cluster with barycenter $\mu_1^*$ than $\mu_4$ coming from a second cluster with barycenter $\mu_2^*$, it will be incorrectly assigned to the second cluster using the centroid-based criterion (7), since $W_2(\mu_3, \mu_1^*) > W_2(\mu_3, \mu_2^*) > \max\{W_2(\mu_3, \mu_1), W_2(\mu_3, \mu_2)\}$. In contrast, for Euclidean spaces due to the following equivalent formulation of the generalized

parallelogram law (5),

$$\sum_{i=1}^{n}\|X - X_i\|_2^2 = n\|X - \bar{X}\|_2^2 + \sum_{i=1}^{n}\|X_i - \bar{X}\|_2^2 \geqslant n\|X - \bar{X}\|_2^2, \quad \text{for any } X \in \mathbb{R}^p,$$

there is always some point $X_{i\dagger}$ satisfying $\|X - X_{i\dagger}\|_2 \geqslant \|X - \bar{X}\|_2$, that is, further away from $X$ than the mean $\bar{X}$; thereby excluding counter-intuitive phenomena as the one shown in Fig. 2.

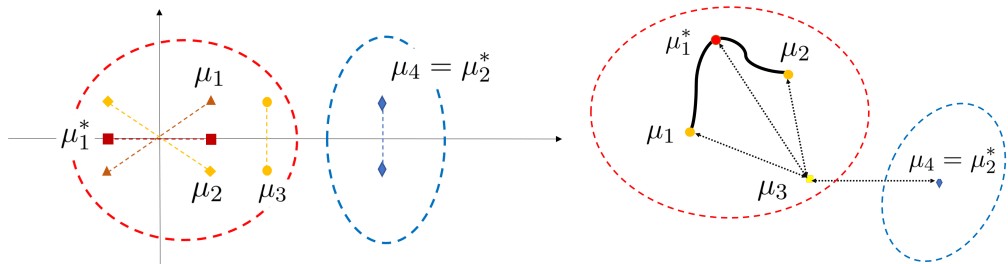

Figure 2: Left: visualization of Example 3 in $\mathbb{R}^2$ and Wasserstein space. Right: the black curve connecting $\mu_1$ and $\mu_2$ depicts the geodesic between them.

Concretely, the first cluster $G_1^*$ is shown in the left panel of Fig. 2 highlighted by a red circle, consisting of $m$ copies of $(\mu_1, \mu_2)$ pairs and one copy of $\mu_3$; the second cluster $G_2^*$ containing copies of $\mu_4$ is highlighted by a blue circle. Each distribution assigns equal probability mass to two points, where the two supporting points are connected by a dashed line for easy illustration. More specifically, we set

$$\mu_1 = 0.5\,\delta_{(x,y)} + 0.5\,\delta_{(-x,-y)}, \quad \mu_2 = 0.5\,\delta_{(x,-y)} + 0.5\,\delta_{(-x,y)},$$

$$\mu_3 = 0.5\,\delta_{(x+\epsilon_1,y)} + 0.5\,\delta_{(x+\epsilon_1,-y)}, \quad \text{and} \quad \mu_4 = 0.5\,\delta_{(x+\epsilon_1+\epsilon_2,y)} + 0.5\,\delta_{(x+\epsilon_1+\epsilon_2,-y)},$$

where $\delta_{(x,y)}$ denotes the point mass measure at point $(x,y)$, and $(x, y, \epsilon_1, \epsilon_2)$ are positive constants. The property of this configuration can be summarized by the following lemma.

*Lemma* 4 (**Configuration characterization**). If $(x, y, \epsilon_1, \epsilon_2)$ satisfies

$$y^2 < \min\{x^2, 0.25\,\Delta_{\epsilon_1,x}\} \quad \text{and} \quad \Delta_{\epsilon_1,x} < \epsilon_2^2 < \Delta_{\epsilon_1,x} + y^2,$$

where $\Delta_{\epsilon_1,x} := \epsilon_1^2 + 2x^2 + 2x\epsilon_1$, then for all sufficiently large $m$ (number of copies of $\mu_1$ and $\mu_2$),

$$W_2(\mu_3, \mu_2^*) < W_2(\mu_3, \mu_1^*) \quad \text{and} \quad \underbrace{\max_{k=1,2} \max_{i,j \in G_k} W_2(\mu_i, \mu_j)}_{\text{largest within-cluster distance}} < \underbrace{\min_{i \in G_1, j \in G_2} W_2(\mu_i, \mu_j)}_{\text{least between-cluster distance}},$$

where $\mu_k^*$ denotes the Wasserstein barycenter of cluster $G_k$ for $k = 1, 2$.

Note that the condition of Lemma 4 implies $y < x$. Therefore, the barycenter between $\mu_1$ and $\mu_2$ is $\tilde{\mu}_1^* := 0.5\,\delta_{(x,0)} + 0.5\,\delta_{(-x,0)}$ lying on the horizontal axis. By increasing $m$, the barycenter $\mu_1^*$ of cluster $G_1^*$ can be made arbitrarily close to $\tilde{\mu}^*$. The second inequality in Lemma 4 shows that all within-cluster distances are strictly less than the between-cluster distances; therefore, clustering based on pairwise distances is able to correctly recover the cluster label of $\mu_3$. However, since $\mu_3$ is closer to the barycenter $\mu_2^*$ of cluster $G_2^*$ according to the first inequality in Lemma 4, it will be mis-classified into $G_2^*$ using the centroid-based criterion. We emphasize that cluster positions in this example are generic and do exist in real data; see Remark 9 and Section 4.3 for further discussions on our experiment results on MNIST data. Moreover, similar to Example 2, a small change in the orientation of distribution $\mu_1$ may completely alter the clustering membership of $\mu_3$ based on the centroid criterion. Specifically, if we slightly increase $x$ to make it exceed $y$, then the barycenter between $\mu_1$ and $\mu_2$ becomes $\bar{\mu}_1^* := 0.5\,\delta_{(0,y)} + 0.5\,\delta_{(0,-y)}$ that lies on the vertical axis. Correspondingly, if based on centroids, then $\mu_3$ should be clustered into $G_1^*$ as it is closer to the barycenter $\mu_1^*$ of $G_1^*$ than the barycenter $\mu_2^*$ of $G_2^*$. Therefore, the centroid-based criterion can be unstable against data perturbations. In comparison, a pairwise distances based criterion always assigns $\mu_3$ into cluster $G_2^*$ no matter $x < y$ or $x > y$. ∎

## 2.2 Clustering based on pairwise distances

Due to the irregularity and non-robustness of centroid-based Wasserstein $K$-means, we instead propose and advocate the use of distance-based Wasserstein $K$-means below, which extends the Euclidean distance-based $K$-means formulation (4) into the Wasserstein space,

$$\min_{G_1,\ldots,G_K} \left\{ \sum_{k=1}^{K} \frac{1}{|G_k|} \sum_{i,j \in G_k} W_2^2(\mu_i, \mu_j) : \bigsqcup_{k=1}^{K} G_k = [n] \right\}. \tag{11}$$

Correspondingly, we can analogously design a greedy algorithm resembling the Wasserstein Lloyd's algorithm described in Section 2.1 that solves the centroid-based Wasserstein $K$-means. Specifically, the greedy algorithm proceeds in an iterative manner as following. Given an initial cluster membership estimate $G_1^{(1)}, \ldots, G_K^{(1)}$, one assigns each probability measure $\mu_1, \ldots, \mu_n$ based on minimizing the averaged squared $W_2$ distances to all current members in every cluster, leading to an updated cluster membership estimate

$$G_k^{(t+1)} = \left\{ i \in [n] : \frac{1}{|G_k^{(t)}|} \sum_{s \in G_k^{(t)}} W_2^2(\mu_i, \mu_s) \leqslant \frac{1}{|G_j^{(t)}|} \sum_{s \in G_j^{(t)}} W_2^2(\mu_i, \mu_s), \quad \forall j \in [K] \right\}. \tag{12}$$

We arbitrarily select among the least $W_2$ distance clusters in the case of a tie. We highlight that the center-based and distance-based Wasserstein $K$-means formulations may not necessarily be equivalent to yield the same cluster labels (cf. Example 3). Below, we shall give some example illustrating connections to the standard $K$-means clustering in the Euclidean space.

*Example* 5 (**Degenerate probability measures**). If the probability measures are Dirac at point $X_i \in \mathbb{R}^p$, i.e., $\mu_i = \delta_{X_i}$, then the Wasserstein $K$-means is the same as the standard $K$-means since $W_2(\mu_i, \mu_j) = \|X_i - X_j\|_2$. ∎

*Example* 6 (**Gaussian measures**). If $\mu_i = N(m_i, V_i)$ with positive-definite covariance matrices $\Sigma_i \succ 0$, then the squared 2-Wasserstein distance can be expressed as the sum of the squared Euclidean distance on the mean vector and

$$d^2(V_i, V_j) = \text{Tr}\left[ V_i + V_j - 2\left(V_i^{1/2} V_j V_i^{1/2}\right)^{1/2}\right], \tag{13}$$

the squared *Bures distance* on the covariance matrix [Bhatia et al., 2019]. Here, we use $V^{1/2}$ to denote the unique symmetric square root matrix of $V \succ 0$. That is,

$$W_2^2(\mu_i, \mu_j) = \|m_i - m_j\|_2^2 + d^2(V_i, V_j). \tag{14}$$

Then the Wasserstein $K$-means, formulated either in (7) or (11), can be viewed as a *covariance-adjusted* Euclidean $K$-means by taking account into the shape or orientation information in the (non-degenerate) Gaussian inputs. ∎

*Example* 7 (**One-dimensional probability measures**). If $\mu_i$ are probability measures on $\mathbb{R}$ with cumulative distribution function (cdf) $F_i$, then the Wasserstein distance can be written in terms of the *quantile transform*

$$W_2^2(\mu_i, \mu_j) = \int_0^1 [F_i^-(u) - F_j^-(u)]^2 \, du, \tag{15}$$

where $F^-$ is the generalized inverse of the cdf $F$ on $[0, 1]$ defined as $F^-(u) = \inf\{x \in \mathbb{R} : F(x) > u\}$ (cf. Theorem 2.18 [Villani, 2003]). Thus the one-dimensional probability measures in Wasserstein space can be isometrically embedded in a flat $L^2$ space, and we can bring back the equivalence of the Wasserstein and Euclidean $K$-means clustering methods. ∎

## 3 SDP relaxation and its theoretic guarantee

Note that Wasserstein Lloyd's algorithm requires to use and compute the barycenter in (7) and (8) at each iteration, which can be computationally expensive when the domain dimension $d$ is large or the configuration is ill-conditioned (cf. Example 2). On the other hand, it is known that solving the distance-based $K$-means (4) is worst-case NP-hard for Euclidean data. Thus we expect solving

the distance-based Wasserstein $K$-means (11) is also computationally hard. A common way is to consider convex relaxations to approximate the solution of (11). It is known that certain SDP relaxation is information-theoretically tight for (4) when the data $X_1, \ldots, X_n \in \mathbb{R}^p$ are generated from a Gaussian mixture model with isotropic known variance [Chen and Yang, 2021]. In this paper, we extend the idea into Wasserstein setting for solving (11).

A typical SDP relaxation for Euclidean data uses pairwise inner products to construct an affinity matrix for clustering [Peng and Wei, 2007]; unfortunately, due to the non-flatness nature, a globally well-defined inner product does not exist for Wasserstein spaces with dimension higher than one. Therefore, we will derive a Wasserstein SDP relaxation to the combinatorial optimization problem (4) using the squared distance matrix $A_{n \times n} = \{a_{ij}\}$ with $a_{ij} = W_2^2(\mu_i, \mu_j)$. Concretely, we can one-to-one reparameterize any partition $(G_1, \ldots, G_K)$ as a binary *assignment matrix* $H = \{h_{ik}\} \in \{0,1\}^{n \times K}$ such that $h_{ik} = 1$ if $i \in G_k$ and $h_{ik} = 0$ otherwise. Then (11) can be expressed as a nonlinear 0-1 integer program,

$$\min \left\{ \langle A, HBH^\top \rangle : H \in \{0,1\}^{n \times K}, H\mathbf{1}_K = \mathbf{1}_n \right\}, \tag{16}$$

where $\mathbf{1}_n$ is the $n \times 1$ vector of all ones and $B = \text{diag}(|G_1|^{-1}, \ldots, |G_K|^{-1})$. Changing of variable to the *membership matrix* $Z = HBH^\top$, we note that $Z_{n \times n}$ is a symmetric positive semidefinite (psd) matrix $Z \succeq 0$ such that $\text{Tr}(Z) = K, Z\mathbf{1}_n = \mathbf{1}_n$, and $Z \geqslant 0$ entrywise. Thus we obtain the SDP relaxation of (11) by only preserving these convex constraints:

$$\min_{Z \in \mathbb{R}^{n \times n}} \left\{ \langle A, Z \rangle : Z^\top = Z, Z \succeq 0, \text{Tr}(Z) = K, Z\mathbf{1}_n = \mathbf{1}_n, Z \geqslant 0 \right\}. \tag{17}$$

To theoretically justify the SDP formulation (17) of Wasserstein $K$-means, we consider the scenario of clustering Gaussian distributions in Example 6, where the Wasserstein distance (14) contains two separate components: the Euclidean distance on mean vector and the Bures distance (13) on covariance matrix. Without loss of generality, we focus on mean-zero Gaussian distributions since optimal separation conditions for exact recovery based on the Euclidean mean component have been established in [Chen and Yang, 2021]. Suppose we observe Gaussian distributions $\nu_i \sim N(0, V_i)$, $i \in [n]$ from $K$ groups $G_1^*, \cdots, G_K^*$, where cluster $G_k^*$ contains $n_k$ members, and the covariance matrices have the following clustering structure: if $i \in G_k^*$, then

$$V_i = (I + tX_i)V^{(k)}(I + tX_i) \quad \text{with } X_1, \ldots, X_n \overset{i.i.d.}{\sim} SymN(0,1), \tag{18}$$

where the psd matrix $V^{(k)}$ is the center of the $k$-th cluster, $SymN(0,1)$ denotes the symmetric random matrix with i.i.d. standard normal entries, and $t$ is a small perturbation parameter such that $(I + tX_i)$ is psd with high probability. For zero-mean Gaussian distributions, we have $W_2(N(0,V), N(0,U)) = d(V,U)$ according to (14). Note that on the Riemannian manifold of psd matrices, the geodesic emanating from $V^{(k)}$ in the direction $X$ as a symmetric matrix can be linearized by $V = (I + tX)V^{(k)}(I + tX)$ in a small neighborhood of $t$, thus motivating the parameterization of our statistical model in (18). The next theorem gives a separation lower bound to ensure exact recovery of the clustering labels for Gaussian distributions.

*Theorem* 8 (**Exact recovery for clustering Gaussians**). Let $\Delta^2 := \min_{k \neq l} d^2(V^{(k)}, V^{(l)})$ denote the minimal pairwise separation among clusters, $\bar{n} := \max_{k \in [K]} n_k$ (and $\underline{n} := \min_{k \in [K]} n_k$) the maximum (minimum) cluster size, and $m := \min_{k \neq l} \frac{2n_k n_l}{n_k + n_l}$ the minimal pairwise harmonic mean of cluster sizes. Suppose the covariance matrix $V_i$ of Gaussian distribution $\nu_i = N(0, V_i)$ is independently drawn from model (18) for $i = 1, 2, \ldots, n$. Let $\beta \in (0,1)$. If the separation $\Delta^2$ satisfies

$$\Delta^2 > \bar{\Delta}^2 := \frac{C_1 t^2}{\min\{(1-\beta)^2, \beta^2\}} \mathcal{V} p^2 \log n, \tag{19}$$

then the SDP (17) achieves exact recovery with probability at least $1 - C_2 n^{-1}$, provided that

$$\underline{n} \geq C_3 \log^2 n, \ \ t \leq C_4 \sqrt{\log n} / \big[(p + \log \bar{n})\mathcal{V}^{1/2} T_v^{1/2}\big], \ \ n/m \leq C_5 \log n,$$

where $\mathcal{V} = \max_k \left\| V^{(k)} \right\|_{\text{op}}, T_v = \max_k \text{Tr}\big[\big(V^{(k)}\big)^{-1}\big]$, and $C_i, i = 1, 2, 3, 4, 5$ are constants.

*Remark* 9 (**Further insight on pitfalls of barycenter-based Wasserstein $K$-means**). Theorem 8 suggests that different from Euclidean data, distributions after centering can be clustered if scales and rotation angles vary (i.e., covariance-adjusted). We further illustrate the rotation and scale effects on

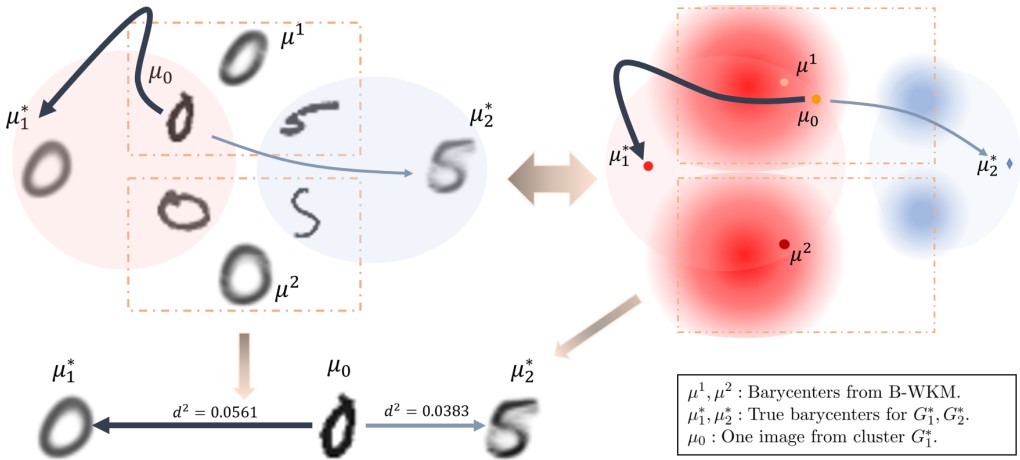

Figure 3: Visualization of misclassification for the barycenter-based Wasserstein $K$-means (B-WKM) on a randomly sampled subset from MNIST (200 digit "0" and 100 digit "5"). The plot at the bottom is a example of misclassified image. The right plot is the abstraction of the images in the Wasserstein space. The color depth indicates the frequency of the distributions. Red and blue colors stand for distributions belong to true clusters "0" and "5".

the MNIST data that may mislead the centroid-based Wasserstein $K$-means, thus providing a real data support for Example 3. Here we randomly sample two unbalanced clusters with 200 numbers of "0" and 100 numbers of "5". Fig. 3 shows the clustering results for the centroid-based Wasserstein $K$-means and its *oracle* version where we replace the estimated barycenters $\mu_1, \mu_2$ with the true barycenters $\mu_1^*, \mu_2^*$ computed on the true labels. Comparing the Wasserstein distances $W_2(\mu_0, \mu_1^*)$ and $W_2(\mu_0, \mu_2^*)$, we see that the image $\mu_0$ (containing digit "0") is closer to $\mu_2^*$ (true barycenter of digit "5") and thus it cannot be classified correctly based on the nearest true barycenter (cf. Fig. 3 on the left). Moreover, Wasserstein $K$-means based on estimated barycenters $\mu_1, \mu_2$ yields two clusters of mixed "0" and "5". In both cases, the misclassification error is characterized by grouping similar degrees of angle and/or stretch. Since there are two highly unbalanced clusters of distributions, Wasserstein $K$-means is likely to enforce larger cluster to separate into two clusters and absorb those around centers (cf. Fig. 3 on the right), leading to larger classification errors. We shall see that in Section 4.3 the distance-based Wasserstein $K$-means and its SDP relaxation have much smaller classification error rate on MNIST for the reason that we explained in Example 3 (cf. Lemma 4). ■

## 4 Experiments

### 4.1 Counter-example in Example 3 revisited

Our first experiment is to back up the claim about the failure of centroid-based Wasserstein $K$-means in Example 3 through simulations. Instead of using point mass measures that may results in instability for computing the barycenters, we use Gaussian distributions with small variance as a smoothed version. We consider $K = 2$, where cluster $G_1^*$ consists of $m_1$ many copies of $(\mu_1, \mu_2)$ pairs and $m_2$ many $\mu_3$, and cluster $G_2^*$ consists of $m_3$ many copies of $\mu_4$. We choose $\mu_i$ as the following two-dimensional mixture of Gaussian distributions $\mu_i = 0.5\, N(a_{i,1}, \Sigma_{i,1}) + 0.5\, N(a_{i,2}, \Sigma_{i,2})$ for $i = 1, 2, 3, 4$. Due to the space limit, detailed simulation setups and parameters are given in Appendix B. From Table 1, we can observe that Wasserstein SDP has achieved exact recovery for all cases while barycenter-based Wasserstein $K$-means has only around $40\%$ exact recovery rate among all repetitions. In addition, Wasserstein SDP is more stable than distance-based Wasserstein $K$-means. Denote $\Delta_k := W^2(\mu_3, \mu_k^*)$ as the squared distance between $\mu_3$ and $\mu_k^*$ for $k = 1, 2$, where $\mu_k^*$ is the barycenter of $G_k^*$. Let $\Delta_* := \max_{k=1,2} \max_{i,j \in G_k} W_2(\mu_i, \mu_j)$ and $\Delta^* := \min_{i \in G_1, j \in G_2} W^2(\mu_i, \mu_j)$ be the maximum within-cluster distance and the minimum between-cluster distance respectively. From Table 7 in the Appendix, we can observe that $\Delta_* < \Delta^*$, from which we can expect Wasserstein SDP to correctly cluster all data points in the Wasserstein space. Moreover, Table 1 shows that about $25\%$ times that the distributions (as $\mu_3$) in $G_1^*$ satisfy $\Delta_1 > \Delta_2$, implying those $\mu_3$ to be likely assigned to the wrong cluster, which is consistent with

Example 3. The experiment results also show that any copy of $\mu_3$ is misclassified whenever exact recovery fails for B-WKM, which means the misclassified rate for $\mu_3$ equals to $(1 - \gamma)$, where $\gamma$ is the exact recovery rate for B-WKM shown in Table 1. Table 6 in the appendix further reports the run time comparison, from which we see that distance-based approaches are more computationally efficient than the barycenter-based one in our settings.

Table 1: Exact recovery rates and frequency of $\Delta_1 > \Delta_2$ for B-WKM among total 50 repetitions in the counter example. W-SDP: Wasserstein SDP, D-WKM: Distance-based Wasserstein $K$-means, B-WKM: Barycenter-based Wasserstein $K$-means. $n$: total number of distributions.

| $n$ | W-SDP | D-WKM | B-WKM | Frequency of $\Delta_1 > \Delta_2$ |
|-----|-------|-------|-------|-----------------------------------|
| 101 | 1.00  | 0.82  | 0.40  | 0.32 |
| 202 | 1.00  | 0.84  | 0.34  | 0.26 |
| 303 | 1.00  | 0.72  | 0.46  | 0.20 |

## 4.2 Gaussian distributions

Next, we simulate random Gaussian measures from model (18) with $K = 4$ and all cluster size equal. We set the centers of each cluster of Gaussians such that all pairwise distances among the barycenters are all equal, i.e., $W_2^2(N(0, V^{(k_1)}), N(0, V^{(k_2)})) \equiv D$ for all $k_1, k_2 \in \{1, 2, 3, 4\}$ with $\mathcal{V} = \max_k \|V^{(k)}\|_{op} \in [4.5, 5.5]$. We fix the dimension $p = 10$ and vary the sample size $n = 200, 400, 600$. And we set the perturbation parameter $t = 10^{-3}$ on the covariance matrix. The simulation results are reported over 100 times in each setting. Fig. 4 shows the misclassification rate (log-scale) versus the squared distance $D$ between centers. We observe that when the distance between centers of clusters are larger than certain threshold (squared distance $D > 10^{-3}$ in this case), then Wasserstein SDP can achieve exact recovery for different $n$, while the misclassification rate for the two Wasserstein $K$-means are stably around $10\%$. And when the distance between centers of clusters are relatively small, the two Wasserstein $K$-means and SDP behave similarly.

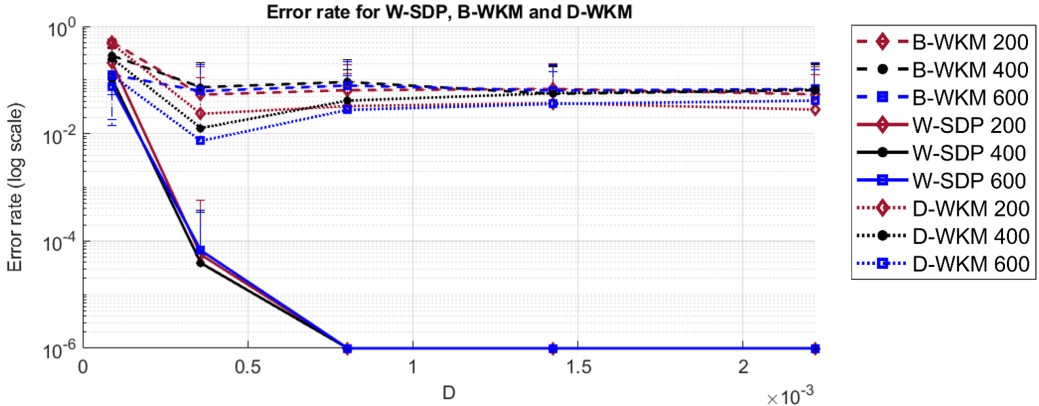

Figure 4: Mis-classification error versus squared distance $D$ from Wasserstein SDP (W-SDP) and barycenter/distance-based Wasserstein $K$-means (B-WKM and D-WKM) for clustering Gaussians under $n \in \{200, 400, 600\}$. Due to the log-scale, $10^{-6}$ corresponds to exact recovery.

## 4.3 Real-data applications

We consider three benchmark image datasets: MNIST, Fashion-MNIST and USPS handwriting digits. Due to complexity issues, we consider subsets of the whole datasets and randomly choose fixed number of images from each clusters based on 10 replicates for each cases. Here we used Sinkhorn divergence to calculate the Wasserstein distance and the Bregman projection with 100 iterations for computing the barycenters, which is efficient and stable for non-degenerate case in practice. For both Wasserstein $K$-means methods, we use the initialization method in analogue to the $K$-means++ for Euclidean data, i.e., the first cluster barycenter is chosen uniformly at random as one of the distributions, after which each subsequent cluster barycenter is

chosen from the remaining distributions with probability proportional to its squared Sinkhorn divergence from the distribution's closest existing cluster barycenter. Codes using MATLAB and Python implementing W-SDP and D-WKM are available at: `https://github.com/Yubo02/Wasserstein-K-means-for-clustering-probability-distributions`.

For MNIST dataset, we choose and fix two clusters: $G_1^*$ containing the number "0" and $G_2^*$ containing the number "5" for two cases. (1) In Case 1, we randomly draw 200 number "0" and 100 number of "5" for each repetition. (2) In Case 2, we double the number and randomly draw 400 number "0" and 200 number of "5" instead. For Fashion-MNIST and USPS handwriting digits dataset, we consider $K = 3$: $G_1^*$ containing the "T-shirt/top" (or the number "0" for USPS handwriting), $G_2^*$ containing the "Trouser" (or the number "5" for USPS handwriting) and $G_3^*$ containing the "Dress" (or the number "7" for USPS handwriting). The cluster sizes are unbalanced where we randomly choose 200, 100 and 100 number from $G_1^*, G_2^*$ and $G_3^*$ respectively. The error rates are shown in Table 2. Detailed setups and more results about $F_1$ scores (analogous to error rates) as well as time costs are placed at Appendix A due to space limit.

From Table 2, we can observe that the performances for Wasserstein SDP (W-SDP) and distance-based Wasserstein $K$-means (D-WKM) are better compared with barycenter-based Wasserstein $K$-means (B-WKM) for all cases. And the results from Table 3 in Appendix A by using $F_1$ score are consistent with the results from Table 2. The original $K$-means method behaves similar as barycenter-based Wasserstein $K$-means in some cases and behaves less preferable for cases such as our experiment on USPS handwriting digits. In particular, the visualization of the clustering results for case 1 has been shown in Fig. 3. From this figure we can find that the classification criterion for B-WKM will end up with the closeness to certain shape of "0", which is characterized by certain angle or the degree of stretch. And this will lead to the high misclassification error for barycenter-based or centroid-based Wasserstein $K$-means.

Table 2: Error rate (SD) for clustering three benchmark datasets: MNIST, Fashion-MNIST and USPS handwriting digits. $MNIST_1$ ($MNIST_2$) refers to the results of Case 1 (Case 2) for MNIST dataset.

|  | W-SDP | D-WKM | B-WKM | KM |
|---|---|---|---|---|
| $MNIST_1$ | 0.235 (0.045) | 0.156 (0.057) | 0.310 (0.069) | 0.295 (0.066) |
| $MNIST_2$ | 0.279 (0.050) | 0.185 (0.097) | 0.324 (0.032) | 0.362 (0.033) |
| Fashion-MNIST | 0.082 (0.020) | 0.056 (0.014) | 0.141 (0.059) | 0.138 (0.099) |
| USPS handwriting | 0.206 (0.020) | 0.159 (0.061) | 0.240 (0.045) | 0.284 (0.025) |

## 5 Discussion

In this paper, we observed and analyzed the peculiar behaviors of Wasserstein barycenters and their results in clustering probability distributions. After that, we proposed the distance-based K-means approach (D-WKW) and its semidefinite program relaxation (W-SDP) by showing the exact recovery results for Gaussians theoretically and numerically. For several real benchmark datasets, we showed results where D-WKM and W-SDP could outperform barycenter-based K-means approach (B-WKM). And we focused on one unbalanced case of two clusters from MNIST and analyzed its behavior through visualization of misclassification for the barycenter-based Wasserstein K-means. The corresponding time costs for B-WKM, D-WKM and W-SDP for benchmark datasets suggest that the scalability for them and especially our approaches could be serious when sample size is large. One of our future goals would be addressing the corresponding computational complexity issues for real data. Another goal is to find out more conclusive results where our approaches are preferable within or out of the realm of unbalanced clusters for real datasets.

## Acknowledgments and Disclosure of Funding

Xiaohui Chen was partially supported by NSF CAREER grant DMS-1752614. Yun Yang was partially supported by NSF grant DMS-2210717.

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
