# A  Additional details on application for real datasets in Section 4.3

In this section, we provide more details of setups and results for real applications in Section 4.3. The results of error rates, $F_1$ scores and time costs are shown in Table 2, Table 3 and Table 4 respectively, which are based on 10 replicates.[1]

Table 3: $F_1$ score (SD) for clustering three benchmark datasets: MNIST, Fashion-MNIST and USPS handwriting digits. $MNIST_1$ ($MNIST_2$) refers to the results of Case 1 (Case 2) for MNIST dataset.

|  | W-SDP | D-WKM | B-WKM | KM |
|---|---|---|---|---|
| $MNIST_1$ | 0.771 (0.044) | 0.842 (0.056) | 0.698 (0.067) | 0.708 (0.063) |
| $MNIST_2$ | 0.729 (0.049) | 0.814 (0.093) | 0.685 (0.031) | 0.647 (0.032) |
| Fashion-MNIST | 0.919 (0.018) | 0.934 (0.036) | 0.817 (0.117) | 0.791(0.168) |
| USPS handwriting | 0.799 (0.019) | 0.835 (0.081) | 0.761 (0.060) | 0.689 (0.093) |

Table 4: Time cost (SD) for clustering three benchmark datasets: MNIST, Fashion-MNIST and USPS handwriting digits. $MNIST_1$ ($MNIST_2$) refers to the results of Case 1 (Case 2) for MNIST dataset.

|  | W-SDP | D-WKM | B-WKM | KM |
|---|---|---|---|---|
| $MNIST_1$ | 525.13 (4.70) | 524.80 (4.92) | 388.87 (647.15) | 0.01 (0.01) |
| $MNIST_2$ | 2187.66 (74.67) | 2160.91 (7.26) | 693.67 (142.57) | 0.02 (0.00) |
| Fashion-MNIST | 849.24 (7.09) | 852.49 (8.60) | 463.28 (176.32) | 0.01 (0.00) |
| USPS handwriting | 1100.87 (19.13) | 1098.05 (16.41) | 317.12 (113.94) | 0.02 (0.01) |

First we run our Wasserstein SDP algorithm against Wasserstein $K$-means on the MNIST dataset for two cases. (1) For the first case, we choose two clusters: $G_1^*$ containing the number "0" and $G_2^*$ containing the number "5", so that the number of clusters is $K = 2$ in the algorithms. The cluster sizes are unbalanced with $|G_1^*|/|G_2^*| = 2$, where we randomly choose 200 number "0" and 100 number of "5" for each repetition. (2) For the second case, we follow the same settings as case 1 except that randomly choose 400 number "0" and 200 number of "5" for each repetition.

Next we considered benchmark dataset Fashion-MNIST $28 \times 28$ containing 10 clusters of $28 \times 28$ greyscale images of clothes. Here we choose three clusters: $G_1$ containing the "T-shirt/top", $G_2$ containing the "Trouser" and $G_3$ containing the "Dress", so that the number of clusters is $K = 3$ in the algorithms. The cluster sizes are unbalanced where we randomly choose $200, 100$ and $100$ number from $G_1, G_2$ and $G_3$ respectively for each repetition.

Finally, we consider the USPS handwriting dataset, analogous to MNIST, which contains digits automatically scanned from envelopes by the U.S. Postal Service containing a total of 9,298 $16 \times 16$ pixel grayscale samples. We choose three clusters: $G_1$ containing the number "0", $G_2$ containing the number "5" and $G_3$ containing the number "7", so that the number of clusters is $K = 3$ in the algorithms. The cluster sizes are unbalanced where we randomly choose $200, 100$ and $100$ number from $G_1, G_2$ and $G_3$ respectively for each repetition.

Now if we look at the time cost for two cases on MNIST ($MNIST_1$ and $MNIST_2$) in Table 4, we can see that Wasserstein SDP (W-SDP), distance-based Wasserstein $K$-means (D-WKM) and barycenter-based Wasserstein $K$-means (B-WKM) all have time complexity issues when we enlarge $n$. The large variance for B-WKM for $MNIST_1$ is due to the convergence of the algorithm. The total iterations for B-WKM in case 1 achieves maximum iteration 100 for 1 replicate out of 10 total replicates. More arguments for time complexity can be found in Appendix B.

# B  Additional details on simulation studies in Section 4.1

In this section, we provide more details of our simulation setups and results for Gaussian mixtures in Section 4.1.

---

[1]We run all the simulations and experiments except for USPS datasets on the machine with Intel Core i7-10700K 3.80 GHz 64 bit 8-core 16 Tread Processor and 16 GB DDR4 Memory; run experiments on USPS datasets with 1.6 GHz Dual-Core Intel Core i5 and 8 GB 2133 MHz LPDDR3 Memory.

We set $m_1 = 40 \cdot r, m_2 = r, m_3 = 20 \cdot r$ which means that there are total $81 \cdot r$ number of distributions in $G_1$, $20 \cdot r$ distributions in $G_2$. The $r$ is set to be $1, 2, 3$, where we have $n = 101, 202, 303$ respectively. The mean for the Gaussian distributions are shown in the table below. The entries of covariance matrices for the Gaussian distributions are chosen to be $O(10^{-3})$ for $\mu_1, \mu_2$ and they are chosen to be $O(10^{-6})$ for $\mu_3$ and $\mu_4$. Then we scale down the distribution with scaling parameter equals 0.5. This ensures that with high probability, all the distributions will fall into the bounded range $[0,1] \times [0,1]$.

The algorithm we use to get the barycenter is Frank-Wolfe algorithm with 200 iterations. And we use Sinkhorn divergence to calculate the Wasserstein distance. The regularization parameters for both algorithms are chosen to be $10^{-3}$. To approximate the true distribution, first we divide $[0,1] \times [0,1]$ range into $80 \times 80$ grids, then we randomly sample 600 samples each time and count the number of times it falls into certain grid to approximate the distribution. The results show us that for each $n$ and each iterations among 50 repetitions, all the distributions in $G_2^*$ will be assigned to same cluster, so it will be reasonable to define that $\mu_3$ is misclassified if any copy of them are in the same cluster of an arbitrarily chosen $\mu$ from $G_2^*$.

The arrangement of mean for Gaussian mixture models shown in Table 5 indicates that the distributions are set based on Example 3. Recall that in Section 4.1, $\Delta_* := \max_{k=1,2} \max_{i,j \in G_k} W_2(\mu_i, \mu_j)$ and $\Delta^* := \min_{i \in G_1, j \in G_2} W^2(\mu_i, \mu_j)$ are the maximum within-cluster distance and the minimum between-cluster distance respectively. Table 7 shows that $\Delta_* < \Delta^*$ on average and $\Delta_* < \Delta^*$ for around $80\%$ among 50 repetitions. So we can expect Wasserstein SDP to correctly cluster all data points in the Wassertein space. From Table 6 we can observe that in our settings the time cost for Wasserstein SDP and distance-based Wasserstein $K$-means is relatively lower than the time cost for barycenter-based Wasserstein $K$-means. But we can see that as $n$ increases, the time cost for B-WKM grows almost linearly w.r.t. $n$ while almost quadratically for W-SDP and D-WKM. Thus we should expect relatively higher time cost for W-SDP and D-WKM when $n$ is sufficiently large, where we can consider several methods to bring down the time cost (e.g., subsampling-based method for SDP from Zhuang et al. [2022]).

Computationally speaking, the calculations of Wasserstein distances and barycenters are usually based on one-step discretization and one-step application of entropic regularization methods such as Sinkhorn (Genevay et al. [2018], Janati et al. [2020]). Dvinskikh and Tiapkin [2020] shows that the complexity of calculating barycenters should be of the order $O(nd^2/\epsilon^2))$ or $O(ng^4/\epsilon^2))$, where $n$ is the total number of distributions, $d = g^2$ is the discretization size, e.g. $g = 28$ for MNIST datasets and $\epsilon$ is the numerical accuracy; while Le et al. [2021] gives a $O(d^2/\epsilon)$ or $O(g^4/\epsilon)$ complexity algorithm for calculating the Wasserstein distance on robust optimal transport.

Table 5: Positions $(x, y) \in \mathbb{R}^2$ of means for two-dimensional mixture of Gaussian distributions for the counter example in Section 4.1.

|   | $a_{1,1}$ | $a_{1,2}$ | $a_{2,1}$ | $a_{2,2}$ | $a_{3,1}$ | $a_{3,2}$ | $a_{4,1}$ | $a_{4,2}$ |
|---|---|---|---|---|---|---|---|---|
| $x$ | 0.75 | 0.25 | 0.75 | 0.25 | 0.9 | 0.9 | 1.3 | 1.3 |
| $y$ | 1.15 | 0.85 | 0.85 | 1.15 | 0.85 | 1.15 | 0.75 | 1.25 |

Table 6: The time cost with standard deviation shown in parentheses for the counter example. TC: Time cost, W-SDP: Wasserstein SDP, D-WKM: Distance-based Wasserstein $K$-means, B-WKM: Barycenter-based Wasserstein $K$-means.

| $n$ | TC for W-SDP (SD) | TC for D-WKM | TC for B-WKM (SD) |
|---|---|---|---|
| 101 | 14.50 (0.5873) | 14.15 (0.5132) | 181.1 (372.4) |
| 202 | 56.94 (1.490) | 54.98 (1.516) | 341.0 (136.2) |
| 303 | 128.4 (3.640) | 123.9 (3.606) | 549.2 (200.2) |

## C  Background on optimal transport

The optimal transport (OT) problem (a.k.a. the Monge problem) is to find an optimal map $T^* : \mathbb{R}^p \to \mathbb{R}^p$ for transporting a source distribution $\mu_0$ to a target distribution $\mu_1$ that minimizes some cost

Table 7: Estimated Wasserstein distances with standard deviation shown in parentheses and frequency of $\Delta^* > \Delta_*$ for the counter example.

| $n$ | $\Delta_*$ | $\Delta^*$ | Frequency of $\Delta_* < \Delta^*$ |
|---|---|---|---|
| 101 | 0.1978 (0.0055) | 0.2046 (0.0050) | 0.8200 |
| 202 | 0.1990 (0.0058) | 0.2050 (0.0051) | 0.8200 |
| 303 | 0.1996 (0.0067) | 0.2052 (0.0050) | 0.7600 |

function $c : \mathbb{R}^p \times \mathbb{R}^p \to \mathbb{R}$:

$$\min_{T:\mathbb{R}^p \to \mathbb{R}^p} \left\{ \int_{\mathbb{R}^p} c(x, T(x)) d\mu_0(x) : T_\sharp \mu_0 = \mu_1 \right\}, \tag{20}$$

where $T_\sharp \mu$ denotes the pushforward measure defined by $(T_\sharp \mu)(B) = \mu(T^{-1}(B))$ for measurable subset $B \subset \mathbb{R}^p$. A standard example of the cost function is the quadratic cost $c(x, y) = \|x - y\|_2^2$. The Monge problem (20) with the quadratic cost induces a metric, known as the *2-Wasserstein distance*, on the space $\mathcal{P}_2(\mathbb{R}^p)$ of probability measures on $\mathbb{R}^p$ with finite second moments. In particular, the 2-Wasserstein distance can be expressed in the relaxed Kantorovich form:

$$W_2^2(\mu_0, \mu_1) := \min_\gamma \left\{ \int_{\mathbb{R}^p \times \mathbb{R}^p} \|x - y\|_2^2 d\gamma(x, y) \right\}, \tag{21}$$

where minimization over $\gamma$ runs over all possible couplings with marginals $\mu_0$ and $\mu_1$ [Villani, 2003]. It is well-known from Brenier's theorem [Brenier, 1991] that if the source measure $\mu_0$ does not charge on small subsets of $\mathbb{R}^p$ (i.e., subsets of Hausdorff dimension at most $p - 1$), then there exists a unique $\mu_0$-almost everywhere OT map $T^*$ solving (20). That is, $T^*_\sharp \mu_0 = \mu_1$ and

$$W_2^2(\mu_0, \mu_1) = \int_{\mathbb{R}^p} \|x - T^*(x)\|_2^2 d\mu_0(x).$$

Let $(\mu_t)_{t=0}^1$ be the constant-speed geodesic connecting $\mu_0, \mu_1 \in \mathcal{P}_2(\mathbb{R}^p)$. Then for any $\nu \in \mathcal{P}_2(\mathbb{R}^p)$ and $t \in [0, 1]$, we have

$$W_2^2(\mu_t, \nu) \geqslant (1 - t) W_2^2(\mu_0, \nu) + t W_2^2(\mu_1, \nu) - t(1 - t) W_2^2(\mu_0, \mu_1). \tag{22}$$

The above semiconcavity inequality (22) can be interpreted as that the Wasserstein space $\mathcal{P}_2(\mathbb{R}^p)$ is a *positive curved* metric space (PC-space) in the sense of Alexandrov (cf. Section 7.3 and Section 12.3 in Ambrosio et al. [2005]).

## D   Additional proofs

In this section, we will give detailed proofs for Example 1, Lemma 4 and Theorem 8. For the proof of Theorem 8, we will first introduce the main part and put the rest proofs of corresponding lemmas at the end of this section to make it clear.

### D.1   Proof of Example 1

Recall that $\mu_0(s) = (1 - s)/2$ and $\mu_1(s) = (1 + s)/2$ are probability densities supported on the line segments $L_0 = \{(s, as) : s \in [-1, 1]\}$ and $L_1 = \{(s, -as) : s \in [-1, 1]\}$ for some $a \in (0, 1)$, respectively. To derive the optimal transport (OT) map $T$ from $\mu_0$ to $\mu_1$, it suffices to consider the one-dimensional OT problem by parameterization of $T : [-1, 1] \to [-1, 1]$ identified via $(s, as) \mapsto (T(s), -aT(s))$. Then our goal is to find the solution to the following optimization problem

$$\min_{T:T_\sharp \mu_0 = \mu_1} \int_{-1}^1 \|(s, -as) - (T(s), -aT(s))\|_2^2 d\mu_0(s)$$

$$= \min_{T:T_\sharp \mu_0 = \mu_1} \int_{-1}^1 [(s - T(s))^2 + a^2(s + T(s))^2] d\mu_0(s)$$

$$= (1 - a^2) \times \min_{T:T_\sharp \mu_0 = \mu_1} \int_{-1}^1 \left[ \sqrt{\frac{1 + a^2}{1 - a^2}} T(s) - s \right]^2 d\mu_0(s) + \text{constant},$$

where the constant does not depend on $T$. Now rescale the distribution density $\mu_1$ to

$$\tilde{\mu}_1(s) = \sqrt{\frac{1-a^2}{1+a^2}}\, \mu_1\left(\sqrt{\frac{1-a^2}{1+a^2}}s\right) \quad \text{for } s \in \left[-\sqrt{\frac{1+a^2}{1-a^2}}, \sqrt{\frac{1+a^2}{1-a^2}}\right],$$

and define the transport map $\tilde{T} = \sqrt{\frac{1+a^2}{1-a^2}}\, T$ on $[-1, 1]$. To find the OT map $T$ such that $T_\sharp \mu_0 = \mu_1$, it suffices to find the OT map $\tilde{T}$ such that $\tilde{T}_\sharp \mu_0 = \tilde{\mu}_1$, i.e.,

$$\min_{\tilde{T}:\tilde{T}_\sharp \mu_0 = \tilde{\mu}_1} \int_{-1}^{1} [\tilde{T}(s) - s]^2 d\mu_0(s),$$

whose solution is known as the *quantile transform* for one-dimensional distributions. Specifically, let

$$F_0(s) = \int_{-1}^{s} \mu_0(t)dt = \frac{1}{2}\left(s - \frac{1}{2}s^2 + \frac{3}{2}\right) \quad \text{for } s \in [-1, 1],$$

be the cumulative distribution function (cdf) of the density $\mu_0$ and

$$\tilde{F}_1(s) = \int_{-\sqrt{\frac{1+a^2}{1-a^2}}}^{s} \tilde{\mu}_1(t)dt = \frac{1}{2}\left(\sqrt{\frac{1-a^2}{1+a^2}}s + \frac{1}{2}\cdot\frac{1-a^2}{1+a^2}s^2 + \frac{1}{2}\right) \quad \text{for } s \in \left[-\sqrt{\frac{1+a^2}{1-a^2}}, \sqrt{\frac{1+a^2}{1-a^2}}\right],$$

be the cdf of the density $\tilde{\mu}_1$. It is easy to find that

$$\tilde{F}_1^{-1}(y) = \sqrt{\frac{1+a^2}{1-a^2}}(\sqrt{4y} - 1) \quad \text{for } y \in [0, 1].$$

Then the OT map $\tilde{T}$ from $\mu_0$ to $\tilde{\mu}_1$ is given by

$$\tilde{T}(s) = \tilde{F}_1^{-1} \circ F_0(s) = \sqrt{\frac{1+a^2}{1-a^2}}[-1 + \sqrt{4 - (1-s)^2}], \quad s \in [-1, 1].$$

This gives the OT map $T$ from $\mu_0$ to $\mu_1$ (in the one-dimensional parameterization form) as

$$T(s) = -1 + \sqrt{4 - (1-s)^2}. \tag{23}$$

Thus, the OT map $T$ from $\mu_0$ to $\mu_1$ as (degenerate) probability distribution in $\mathbb{R}^2$ is given by

$$T(s, as) = \left(-1 + \sqrt{4 - (1-s)^2},\; -a \cdot (-1 + \sqrt{4 - (1-s)^2})\right).$$

## D.2 Proof of Lemma 4 in Section 2.1

Recall the settings as following

$$\mu_1 = 0.5\,\delta_{(x,y)} + 0.5\,\delta_{(-x,-y)}, \quad \mu_2 = 0.5\,\delta_{(x,-y)} + 0.5\,\delta_{(-x,y)},$$
$$\mu_3 = 0.5\,\delta_{(x+\epsilon_1,y)} + 0.5\,\delta_{(x+\epsilon_1,-y)}, \quad \text{and} \quad \mu_4 = 0.5\,\delta_{(x+\epsilon_1+\epsilon_2,y)} + 0.5\,\delta_{(x+\epsilon_1+\epsilon_2,-y)},$$

where $\delta_{(x,y)}$ denotes the point mass measure at point $(x, y) \in \mathbb{R}^2$, and $(x, y, \epsilon_1, \epsilon_2)$ are positive constants.

*Lemma 4* (**Configuration characterization**). If $(x, y, \epsilon_1, \epsilon_2)$ satisfies

$$y^2 < \min\{x^2, 0.25\,\Delta_{\epsilon_1,x}\} \quad \text{and} \quad \Delta_{\epsilon_1,x} < \epsilon_2^2 < \Delta_{\epsilon_1,x} + y^2,$$

where $\Delta_{\epsilon_1,x} := \epsilon_1^2 + 2x^2 + 2x\epsilon_1$, then for all sufficiently large $m$ (number of copies of $\mu_1$ and $\mu_2$),

$$W_2(\mu_3, \mu_2^*) < W_2(\mu_3, \mu_1^*) \quad \text{and} \quad \underbrace{\max_{k=1,2}\max_{i,j\in G_k} W_2(\mu_i, \mu_j)}_{\text{largest within-cluster distance}} < \underbrace{\min_{i\in G_1, j\in G_2} W_2(\mu_i, \mu_j)}_{\text{least between-cluster distance}},$$

where $\mu_k^*$ denotes the Wasserstein barycenter of cluster $G_k$ for $k = 1, 2$.

*Proof.* For any $w_i \in \mathbb{R}^2, i = 1, 2, 3, 4$, let $\mu = 0.5\,\delta_{w_1} + 0.5\,\delta_{w_2}$, $\nu = 0.5\,\delta_{w_3} + 0.5\,\delta_{w_4}$. By definition of Wasserstein distance we can show that

$$W_2^2(\mu, \nu) = 0.5\min\{\|w_1 - w_3\|^2 + \|w_2 - w_4\|^2, \|w_1 - w_4\|^2 + \|w_2 - w_3\|^2\}.$$

Let $\mu_0 = 0.5\,\delta_{(x,0)} + 0.5\,\delta_{(-x,0)}$, by algebraic calculation it is direct to check

$$W_2(\mu_3, \mu_2^*) < W_2(\mu_3, \mu_0) \qquad \text{and} \qquad \underbrace{\max_{k=1,2}\max_{i,j\in G_k} W_2(\mu_i, \mu_j)}_{\text{largest within-cluster distance}} < \underbrace{\min_{i\in G_1, j\in G_2} W_2(\mu_i, \mu_j)}_{\text{least between-cluster distance}},$$

once plugging in the assumptions. So we only need to show that $\forall \varepsilon, \exists M$, s.t. when $m > M$ we have $W_2^2(\mu_3, \mu_1^*) \geq W_2^2(\mu_3, \mu_0) - \varepsilon$. For notation simplicity, let $v_x = (x,0), v_{-x} = (-x,0), v_1 = (x,y), v_2 = (-x,-y), v_3 = (x,-y), v_4 = (x,-y)$. By definition we know there exist measures $\xi_i, i = 1, 2, 3, 4$, s.t.

$$W_2^2(\mu_1^*, \mu_1) = \int \|v - v_1\|^2 d\xi_1(v) + \int \|v - v_2\|^2 d\xi_2(v),$$

$$W_2^2(\mu_1^*, \mu_2) = \int \|v - v_3\|^2 d\xi_3(v) + \int \|v - v_4\|^2 d\xi_4(v),$$

where $\mu_1^* = \xi_1 + \xi_2 = \xi_3 + \xi_4$ with $\xi_i(\mathbb{R}^2) = 0.5, \forall i$. Furthermore, if we define $\xi_{i,j} = \xi_i \cdot \xi_j / \mu_1^*, i \in \{1,2\}, j \in \{3,4\}$, then $\xi_i = \xi_{i,3} + \xi_{i,4}, \xi_j = \xi_{1,j} + \xi_{2,j}, i \in \{1,2\}, j \in \{3,4\}$. Thus

$$W_2^2(\mu_1^*, \mu_1) + W_2^2(\mu_1^*, \mu_2) = \sum_{i=1}^4 \int \|v - v_i\|^2 d\xi_i(v)$$

$$= \sum_{i\in\{1,2\}, j\in\{3,4\}} \int \|v - v_i\|^2 + \|v - v_j\|^2 d\xi_{i,j}(v).$$

Now suppose $t = \|v - v_x\|$, by algebraic calculation we can get

$$\|v - v_1\|^2 + \|v - v_3\|^2 = t^2 + 2y^2.$$

Choose $T > 0$ s.t. $T^2 < \min\{2x^2 - 2y^2, y^2\}$, then we have

$$W_2^2(\mu_1^*, \mu_1) + W_2^2(\mu_1^*, \mu_2) = \sum_{i\in\{1,2\}, j\in\{3,4\}} \int \|v - v_i\|^2 + \|v - v_j\|^2 d\xi_{i,j}(v)$$

$$\leq \int_{B_T(v_x)} \|v - v_1\|^2 + \|v - v_3\|^2 d\xi_{1,3}(v) + \int_{B_T(v_{-x})} \|v - v_2\|^2 + \|v - v_4\|^2 d\xi_{2,4}(v)$$

$$+ (T^2 + 2y^2)(1 - \xi_{1,3}(B_T(v_x)) - \xi_{2,4}(B_T(v_{-x})))$$

$$= \int_{B_T(v_x)} t_1(v)^2 d\xi_{1,3}(v) + \int_{B_T(v_{-x})} t_2(v)^2 d\xi_{2,4}(v) + 2y^2 + T^2(1 - \xi_{1,3}(B_T(v_x)) - \xi_{2,4}(B_T(v_{-x}))),$$

where $B_t(v)$ stands for the ball with radius $t$ centered at $v$, $t_1(v) := \|v - v_x\|, t_2(v) := \|v - v_{-x}\|$. On the other hand, by definition we know that

$$m \cdot W_2^2(\mu_1^*, \mu_1) + m \cdot W_2^2(\mu_1^*, \mu_2) + W_2^2(\mu_1^*, \mu_3)$$
$$\leq m \cdot W_2^2(\mu_0, \mu_1) + m \cdot W_2^2(\mu_0, \mu_2) + W_2^2(\mu_0, \mu_3)$$
$$= m \cdot (2y^2) + C,$$

where $C := W_2^2(\mu_0, \mu_3)$. So we have $W_2^2(\mu_1^*, \mu_1) + W_2^2(\mu_1^*, \mu_2) \leq 2y^2 + C/m$. i.e.,

$$\int_{B_T(v_x)} t_1(v)^2 d\xi_{1,3}(v) + \int_{B_T(v_{-x})} t_2(v)^2 d\xi_{2,4}(v) + T^2(1 - \xi_{1,3}(B_T(v_x)) - \xi_{2,4}(B_T(v_{-x}))) \leq \frac{C}{m}.$$

So we have

$$\int_{B_T(v_x)} t_1(v)^2 d\xi_{1,3}(v) \leq \frac{C}{m}, \quad \int_{B_T(v_{-x})} t_2(v)^2 d\xi_{2,4}(v) \leq \frac{C}{m},$$

$$0.5 - \xi_{1,3}(B_T(v_x)) \leq \frac{C}{T^2 m}, \quad 0.5 - \xi_{2,4}(B_T(v_{-x})) \leq \frac{C}{T^2 m}.$$

Now suppose $v_{\epsilon_1} := (x + \epsilon_1, y), v_{-\epsilon_1} := (x + \epsilon_1, -y)$, note that $T^2 < y^2 < \epsilon_1^2 + y^2$ and $W_2^2(\mu_3, \mu_0) = 0.5\|v_x - v_{\epsilon_1}\|^2 + 0.5\|v_{-x} - v_{-\epsilon_1}\|^2$. By definition of Wasserstein distance and symmetry we have

$$W_2^2(\mu_3, \mu_1^*) \geq \int_{B_T(v_x)} (\|v_x - v_{\epsilon_1}\| - t_1(v))^2 d\xi_{1,3}(v) + \int_{B_T(v_{-x})} (\|v_{-x} - v_{\epsilon_1}\| - t_2(v))^2 d\xi_{2,4}(v)$$

$$\geq \|v_x - v_{\epsilon_1}\|^2 \xi_{1,3}(B_T(v_x)) + \|v_{-x} - v_{\epsilon_1}\|^2 \xi_{2,4}(B_T(v_{-x}))$$

$$- 2\|v_x - v_{\epsilon_1}\| \int_{B_T(v_x)} t_1(v) d\xi_{1,3}(v) - 2\|v_x - v_{-\epsilon_1}\| \int_{B_T(v_{-x})} t_2(v) d\xi_{2,4}(v)$$

$$\geq W_2^2(\mu_3, \mu_0) - C_1^2 \cdot \frac{C}{T^2 m} - C_2^2 \cdot \frac{C}{T^2 m}$$

$$- 2C_1 \int_{B_T(v_x)} t_1(v) d\xi_{2,4}(v) - 2C_2 \int_{B_T(v_{-x})} t_2(v) d\xi_{2,4}(v),$$

where $C_1 = \|v_x - v_{\epsilon_1}\|, C_2 = \|v_{-x} - v_{\epsilon_1}\|$. Set $\forall \varepsilon > 0$. Finally, by Hölder's inequality we have

$$W_2^2(\mu_3, \mu_1^*) \geq W_2^2(\mu_3, \mu_0) - C_1^2 \cdot \frac{C}{T^2 m} - C_2^2 \cdot \frac{C}{T^2 m}$$

$$- 2C_1 \sqrt{\int_{B_T(v_x)} t_1^2(v) d\xi_{2,4}(v)} - 2C_2 \sqrt{\int_{B_T(v_{-x})} t_2^2(v) d\xi_{2,4}(v)}$$

$$\geq W_2^2(\mu_3, \mu_0) - C_1^2 \cdot \frac{C}{T^2 m} - C_2^2 \cdot \frac{C}{T^2 m} - 2C_1 \sqrt{\frac{C}{m}} - 2C_2 \sqrt{\frac{C}{m}}$$

$$\geq W_2^2(\mu_3, \mu_0) - \varepsilon,$$

for large $m$, as desired. ∎

### D.3 Proof of Theorem 8 in Section 3

*Theorem 8* (**Exact recovery for clustering Gaussians**). Let $\Delta^2 := \min_{k \neq l} d^2(V^{(k)}, V^{(l)})$ denote the minimal pairwise separation among clusters, $\bar{n} := \max_{k \in [K]} n_k$ (and $\underline{n} := \min_{k \in [K]} n_k$) the maximum (minimum) cluster size, and $m := \min_{k \neq l} \frac{2 n_k n_l}{n_k + n_l}$ the minimal pairwise harmonic mean of cluster sizes. Suppose the covariance matrix $V_i$ of Gaussian distribution $\nu_i = N(0, V_i)$ is independently drawn from model (18) for $i = 1, 2, \ldots, n$. Let $\beta \in (0, 1)$. If the separation $\Delta^2$ satisfies

$$\Delta^2 > \bar{\Delta}^2 := \frac{C_1 t^2}{\min\{(1-\beta)^2, \beta^2\}} \mathcal{V} p^2 \log n,$$

then the SDP (17) achieves exact recovery with probability at least $1 - C_2 n^{-1}$, provided that

$$\underline{n} \geq C_3 \log^2 n, \quad t \leq C_4 \sqrt{\log n} / \big[(p + \log \bar{n}) \mathcal{V}^{1/2} T_v^{1/2}\big], \quad n/m \leq C_5 \log n,$$

where $\mathcal{V} = \max_k \big\|V^{(k)}\big\|_{\text{op}}$, $T_v = \max_k \text{Tr}\big[(V^{(k)})^{-1}\big]$, and $C_i, i = 1, 2, 3, 4, 5$ are constants.

*Lemma* 10 (**Dual argument for SDP** (Section B in Chen and Yang [2021])). The sufficient condition for $Z^* = \sum_{k \in [K]} \frac{1}{n_k} 1_{G_k} 1_{G_k}^T$ to be the unique solution of the SDP problem is to find $(\lambda, \alpha, B)$ s.t.

$$(C_1) \quad B \geq 0 \ (B_{G_k G_k} = 0, B_{G_k G_l} > 0, \forall k \neq l),$$

$$(C_2) \quad W_n := \lambda Id + \frac{1}{2}(1\alpha^T + \alpha 1^T) - A - B \succeq 0,$$

$$(C_3) \quad \text{Tr}(W_n Z^*) = 0,$$

$$(C_4) \quad \text{Tr}(B Z^*) = 0,$$

which implies that

$$\alpha_{G_k} = \frac{2}{n_k} A_{G_k G_k} 1_{n_k} - \frac{\lambda}{n_k} 1_{n_k} - \frac{1}{n_k^2} (1_{n_k}^T A_{G_k G_k} 1_{n_k}).$$

$$[B_{G_l G_k} 1_{n_k}]_j = -\frac{n_l + n_k}{2n_l}\lambda + \frac{n_k}{2}\left[\frac{1}{n_l^2}\sum_{s,r\in G_l}d^2(V_s,V_r) - \frac{1}{n_k^2}\sum_{s,r\in G_k}d^2(V_s,V_r)\right]$$
$$+ n_k\left[\frac{1}{n_k}\sum_{r\in G_k}d^2(V_j,V_r) - \frac{1}{n_l}\sum_{r\in G_l}d^2(V_j,V_r)\right],$$

for $k \neq l$, $j \in G_l$.

*Remarks.* It can be justified that if we can find $(\lambda, B)$ satisfying above equations, then $(C_3), (C_4)$ will hold automatically. Details can be found in Section B in Chen and Yang [2021].

Now we will proof the main theorem by two steps. First we will provide a lower bound for $[B_{G_l G_k} 1_{n_k}]_j$. Similar to the argument from Chen and Yang [2021], we want to set $\lambda$ properly such that $(C_1)$ can hold. In the next step we will try to verify that the choice of $(\lambda, \alpha, B)$ and the conditions on the signals could actually imply $(C_2)$. And since number of clusters $K$ is treated as fixed for most practical settings, we will not emphasize $K = O(1)$.

### D.3.1 Proof of main result.

*Step 1* (**Construct** $(\lambda, B)$). Recall $[B_{G_l G_k} 1_{n_k}]_j = -\frac{n_l + n_k}{2n_l}\lambda + n_k L$, where $L$ equals

$$\frac{1}{2}\left[\frac{1}{n_l^2}\sum_{s,r\in G_l}d^2(V_s,V_r) - \frac{1}{n_k^2}\sum_{s,r\in G_k}d^2(V_s,V_r)\right] + \left[\frac{1}{n_k}\sum_{r\in G_k}d^2(V_j,V_r) - \frac{1}{n_l}\sum_{r\in G_l}d^2(V_j,V_r)\right].$$

For $L$ defined above, by Lemma 14, we have

$$L \geq d^2(V^{(l)}, V^{(k)}) - d(V^{(l)}, V^{(k)})K_1 - K_2,$$

w.p. at least $(1 - c/n^2)$, where

$$K_1 = C\sqrt{\log n}t\mathcal{V}^{1/2} + Ct^2(p + \log\bar{n})\mathcal{V}T_v^{1/2},$$
$$K_2 = Ct^2 p^2 \log n\mathcal{V},$$

for some constant $C, c$. Now we chose $\beta \in (0, 1)$ and let $m := \min_{k\neq l}\frac{2n_k n_l}{n_k + n_l}$. If we suppose

$$\Delta \geq Ctp\sqrt{\log n}\mathcal{V}^{1/2}/(1-\beta), \quad t \leq C'\sqrt{\log n}/[(p + \log\bar{n})\mathcal{V}^{1/2}T_v^{1/2}],$$

for some constant $C, C'$, then we have

$$(1 - \beta)d^2(V^{(l)}, V^{(k)}) - d(V^{(l)}, V^{(k)})K_1 - K_2 \geq 0, \forall k \neq l,$$

which implies that

$$L \geq \beta d^2(V^{(l)}, V^{(k)}).$$

Define for $k \neq l$,

$$c_j^{(k,l)} := [B_{G_l G_k} 1_{n_k}]_j, \ j \in G_l,$$
$$r_i^{(k,l)} := [1_{n_l}^T B_{G_l G_k}]_i, \ i \in G_k,$$
$$t^{(k,l)} := 1_{n_l}^T B_{G_l G_k} 1_{n_k},$$
$$(B_{G_l G_k}^{\#})_{ij} := r_i^{(k,l)} c_j^{(k,l)}/t^{(k,l)}.$$

And define $(B_{G_l G_l}^{\#})_{ij} := 0, \forall l$. By setting $\lambda = \frac{\beta}{4}m\Delta^2$, further we have

$$c_j^{(k,l)} \geq \frac{\beta}{2}n_k d^2(V^{(l)}, V^{(k)}), r_i^{(k,l)} \geq \frac{\beta}{2}n_l d^2(V^{(l)}, V^{(k)}), t^{(k,l)} \geq \frac{\beta}{2}n_l n_k d^2(V^{(l)}, V^{(k)}),$$

which implies that $(B_{G_l G_k}^{\#})_{ij} > 0, \forall i \in G_k, j \in G_l$. And $[B_{G_l G_k} 1_{n_k}]_j = [B_{G_l G_k}^{\#} 1_{n_k}]_j$, which means we can construct $B^{\#}$ based on $[B_{G_l G_k} 1_{n_k}]_j$ with $[B_{G_l G_k} 1_{n_k}]_j = [B_{G_l G_k}^{\#} 1_{n_k}]_j$. So essentially, they are the same in the sense that we only care about they quantity through $[B_{G_l G_k} 1_{n_k}]_j$. And thus for notation simplicity, we will use the symbol $B$ instead of $B^{\#}$.

*Step 2* (**Verify the condition for $W_n$ in $(C_2)$**). Next we would like to find sufficient condition for $(C_2)$, i.e.,

$$v^T W_n v \geq 0, \forall v \in \Gamma_K := \text{span}\{1_{G_k} : k \in [K]\}^\perp, \|v\| = 1.$$

Note that $v^T W_n v = \lambda - v^T A v - v^T B v \geq \lambda - v^T B v$. And by definition as well as simple calculation we have

$$v^T B v = \sum_{k=1}^{K} \sum_{l \neq k} \frac{1}{t^{(k,l)}} \left( \sum_{i \in G_k} v_i r_i^{(k,l)} \right) \left( \sum_{j \in G_l} v_j c_j^{(k,l)} \right),$$

$$\sum_{j \in G_l} v_j c_j^{(k,l)} = n_k \sum_{j \in G_l} \left( \frac{1}{n_k} \sum_{r \in G_k} d^2(V_j, V_r) - \frac{1}{n_l} \sum_{r \in G_l} d^2(V_j, V_r) \right) v_j.$$

Further note that

$$\frac{1}{n_k} \sum_{r \in G_k} d^2(V_j, V_r) - \frac{1}{n_l} \sum_{r \in G_l} d^2(V_j, V_r) = d^2(V^{(l)}, V^{(k)}) + E_j^{(k,l)},$$

where

$$E_j^{(k,l)} = \left[ \frac{1}{n_k} \sum_{r \in G_k} d^2(V_j, V_r) - d^2(V_j, V^{(k)}) \right] + \left[ d^2(V_j, V^{(k)}) - d^2(V^{(l)}, V^{(k)}) \right]$$

$$- \frac{1}{n_l} \sum_{r \in G_l} d^2(V_j, V_r).$$

Then by triangle inequality and throwing away the last term of $E_j^{(k,l)}$, we have

$$\sum_{j \in G_l} v_j c_j^{(k,l)} = n_k \sum_{j \in G_l} E_j^{(k,l)} v_j \leq n_k \sum_{j \in G_l} (E_{1,j}^{(k,l)} + E_{2,j}^{(k,l)})|v_j|,$$

where

$$E_{1,j}^{(k,l)} = \frac{1}{n_k} \sum_{r \in G_k} d^2(V^{(k)}, V_r) + \left[ \frac{2}{n_k} \sum_{r \in G_k} d(V^{(k)}, V_r) d(V_j, V^{(k)}) \right],$$

$$E_{2,j}^{(k,l)} = d^2(V^{(l)}, V_j) + 2d(V^{(l)}, V_j) d(V^{(l)}, V^{(k)}).$$

If we set $\tilde{E}_{h,j}^{(k,l)} = E_{h,j}^{(k,l)}/d(V^{(l)}, V^{(k)}), h = 1, 2$, then the inequality can be written as

$$\sum_{j \in G_l} v_j c_j^{(k,l)} \leq n_k d(V^{(l)}, V^{(k)}) \sum_{j \in G_l} (\tilde{E}_{1,j}^{(k,l)} + \tilde{E}_{2,j}^{(k,l)})|v_j|.$$

By Lemma 15 we know

$$\sum_{j \in G_l} \tilde{E}_{1,j}^{(k,l)}|v_j| \leq C\mathcal{V}^{1/2} pt\sqrt{n_l} \left( \sum_{j \in G_l} v_j^2 \right)^{1/2},$$

w.p. $\geq 1 - cn^{-2}$. And by Lemma 16 we have

$$\sum_{j \in G_l} \tilde{E}_{2,j}^{(k,l)}|v_j| \leq Ct\mathcal{V}^{1/2} p(\sqrt{n_l} + \log^2(n)) \left( \sum_{j \in G_l} v_j^2 \right)^{1/2},$$

w.p. $\geq 1 - cn^{-1}$, for some constants $C, c$. Now if we assume $\min_k n_k \geq C \log^2 n$ and notice that $t^{(k,l)} \geq \frac{\beta}{2} n_l n_k d^2(V^{(l)}, V^{(k)})$, then further we can get

$$v^T B v \leq \sum_{k,l} \frac{n_k n_l}{t^{(k,l)}} \sqrt{n_l} \sqrt{n_k} \left( \sum_{j \in G_l} v_j^2 \right)^{1/2} \left( \sum_{i \in G_k} v_i^2 \right)^{1/2} Ct^2 \mathcal{V} p^2$$

$$\leq \frac{Ct^2}{\beta} \left( \sum_l \sum_{j \in G_l} v_j^2 \right)^{1/2} \left( \sum_l n_l \right)^{1/2} \left( \sum_k \sum_{i \in G_k} v_i^2 \right)^{1/2} \left( \sum_k n_k \right)^{1/2} \mathcal{V} p^2$$

$$= \frac{Ct^2}{\beta} p^2 n \mathcal{V},$$

where the second inequality comes from Cauchy-Schwarz inequality. So by assuming

$$\Delta^2 \geq \frac{Ct^2}{\beta^2} \mathcal{V} \cdot p^2 n/m,$$

for some constant $C$, we have

$$v^T W_n v \geq \lambda - v^T B v \geq \frac{\beta}{4} m \Delta^2 - \frac{Ct^2}{\beta} p^2 n \mathcal{V} > 0.$$

Or it is sufficient to assume

$$\Delta^2 \geq \frac{Ct^2}{\beta^2} \mathcal{V} \cdot p^2 \log n,$$

if $n/m = O(\log n)$. To sum up, if we assume

$$\Delta^2 \geq \frac{Ct^2}{\min\{1-\beta,\beta\}^2} \mathcal{V} \cdot p^2 \log n,$$

then w.p. $\geq 1 - c/n$, we have $(C_1) - (C_4)$ hold by the construction of $(\lambda, B)$ for some constants $C, c$. Finally by Lemma 10 we know the solution of SDP $Z^*$ exists uniquely, which is

$$Z^* = \sum_{k \in [K]} \frac{1}{n_k} 1_{G_k} 1_{G_k}^T$$

as desired. ∎

*Remarks.* In our theorem, we focus on the relation between minimum cluster distance $\bar{\Delta}$ with number of distributions $n$, which should be tight enough in the sense that $\bar{\Delta} \asymp \sqrt{\log n}$. This is the same order for the cut-off of exact recovery of SDP for Euclidean case from Chen and Yang [2021].

On the other hand, one sufficient condition for $V_i, i = 1, \ldots, n$ to be psd is $1 - t \max_i \|X_i\|_{op} > 0$, which will hold w.p. $\geq 1 - c/n^2$ if $t \leq C/[\sqrt{p} + \sqrt{\log n}]$ for some constant $C, c$. Recall from our assumption,

$$t \leq c\sqrt{\log n}/[(p + \log \bar{n})\mathcal{V}^{1/2} T_v^{1/2}] \leq c\sqrt{\log n}/(p + \log \bar{n}),$$

for some constant, since $T_v = \max_k \text{Tr}((V^{(k)})^{-1}) \geq p/\min_k \|V^{(k)}\|_{op}$. This indicates that our bound for $t$ guarantees $V_i$ to be psd w.p. $\geq 1 - c/n^2$ as $n \asymp \bar{n}$. One may apply triangle inequality directly to Lemma 14 to get the upper bound of $t$ with less order in $p$, which is of less concern in our theorem, where we put more emphasis on the order in $n$.

### D.3.2 Proofs of lemmas.

Before proving Lemma 14, let us first look at the Taylor expansion for psd matrix.

*Lemma* 11 (**Taylor expansion for psd matrix** (Theorem 1.1 in Del Moral and Niclas [2017])). The square root function $\varphi : Q \in \mathcal{S}_r^+ \mapsto Q^{1/2}$ is Fréchet differentiable at any order on $\mathcal{S}_r^+$ with the first order derivative given for any $(A, H) \in \mathcal{S}_r^+ \times \mathcal{S}_r$ by the formula

$$\nabla \varphi(A) \cdot H = \int_0^\infty e^{-t\varphi(A)} H e^{-t\varphi(A)} dt,$$

where $\mathcal{S}_r^+, \mathcal{S}_r$ are the positive semi-definite matrix and symmetric matrix respectively. The higher order derivatives are defined inductively for any $n \geq 2$ by

$$\nabla^n \varphi(A) \cdot H = -\nabla \varphi(A) \cdot \left[ \sum_{p+q=n-2 \& p,q \geq 0} \frac{n!}{(p+1)!(q+1)!} [\nabla^{p+1} \varphi(A) \cdot H][\nabla^{q+1} \varphi(A) \cdot H] \right].$$

Again from the same paper, we have the Taylor expansion for $\varphi(A)$:

$$\varphi(A + H) = \varphi(A) + \sum_{1 \leq k \leq n} \frac{1}{k!} \nabla^k \varphi(A) \cdot H + \bar{\nabla}^{n+1} \varphi[A, H],$$

with

$$\bar{\nabla}^{n+1} \varphi[A, H] := \frac{1}{n!} \int_0^1 (1-\epsilon)^n \nabla^{n+1} \varphi(A + \epsilon H) \cdot H d\epsilon.$$

*Corollary* 12 (**Decomposition of Wasserstein distance for Gaussians**). If we choose $n = 1$ in Lemma 11, we have for $k \neq l, j \in G_l^*$, and under the assumptions in the Theorem, the following expansion holds.

$$d^2(V_j, V^{(k)}) - d^2(V_j, V^{(l)})$$
$$= d^2(V^{(l)}, V^{(k)}) + \left\langle \mathcal{A}(V^{(l)}, V^{(k)}), t(X_j V^{(l)} + V^{(l)} X_j) + t^2 X_j V^{(l)} X_j \right\rangle$$
$$- d^2(V_j, V^{(l)}) - \Delta_0,$$

$$\frac{1}{n_k} \sum_{r \in G_k} d^2(V_j, V_r) - d^2(V_j, V^{(k)})$$
$$= \left\langle \mathcal{A}(V_j, V^{(k)}), \frac{1}{n_k} \sum_{r \in G_k} t(X_r V^{(k)} + V^{(k)} X_r) + t^2 X_r V^{(k)} X_r \right\rangle - \Delta_1,$$

where $\mathcal{A}(U, V) := Id - U^{1/2}(U^{1/2} V U^{1/2})^{-1/2} U^{1/2}$, for $U, V$ : psd. And $\Delta_0 \leq 0, \Delta_1 \leq 0$, which are extra terms (high order terms in Lemma 11).

*Proof.* By definition we know $d^2(V, U) = W_2^2(\nu, \mu)$, where $\nu \sim N(0, V), \mu \sim N(0, U)$. Thus

$$d^2(V, U) = \text{Tr}(V) + \text{Tr}(U) - 2\text{Tr}[\sqrt{V^{1/2} U V^{1/2}}].$$

So we have

$$d^2(V_j, V^{(k)}) - d^2(V^{(k)}, V^{(l)})$$
$$= \text{Tr}[V_j - V^{(l)}] - 2\text{Tr}\left[\sqrt{(V^{(k)})^{1/2} V_j (V^{(k)})^{1/2}} - \sqrt{(V^{(k)})^{1/2} V^{(l)} (V^{(k)})^{1/2}}\right].$$

On the other hand, by definition we know $V_j = (I + tX_j)V^{(l)}(I + tX_j) = V^{(l)} + t(X_j V^{(l)} + V^{(l)} X_j) + t^2 X_j V^{(l)} X_j$. Then by Lemma 11 and note the second order remainder term is always negative semi-definite, we can directly get the results by first order Taylor expansion. ∎

*Lemma* 13 (**Norm for operator $\mathcal{A}$**). We conclude that for any $U, V$: psd, we have

$$\|\mathcal{A}(U, V) \cdot V^{1/2}\|_F^2 = \|V^{1/2} \cdot \mathcal{A}(U, V)\|_F^2 = d^2(U, V).$$

*Proof.* Suppose we have the SVD

$$U^{1/2} V^{1/2} = Q_1^T \Sigma Q_2,$$

then we have

$$\mathcal{A}(U, V) \cdot V^{1/2} = (I - U^{1/2}(U^{1/2} V U^{1/2})^{-1/2}) V^{1/2}$$
$$= V^{1/2} - U^{1/2} Q_1^T Q_2,$$

which implies that

$$\|\mathcal{A}(U, V) \cdot V^{1/2}\|_F^2 = \text{Tr}(V) + \text{Tr}(U) - 2\text{Tr}(V^{1/2} U^{1/2} Q_1^T Q_2)$$
$$= \text{Tr}(V) + \text{Tr}(U) - 2\text{Tr}(Q_2^T \Sigma Q_2)$$
$$= \text{Tr}(V) + \text{Tr}(U) - 2\text{Tr}(\sqrt{U^{1/2} V U^{1/2}}).$$

∎

*Lemma* 14 (**Lower bound for $L$**). Recall that $L$ equals

$$\frac{1}{2}\left[\frac{1}{n_l^2} \sum_{s,r \in G_l} d^2(V_s, V_r) - \frac{1}{n_k^2} \sum_{s,r \in G_k} d^2(V_s, V_r)\right] + \left[\frac{1}{n_k} \sum_{r \in G_k} d^2(V_j, V_r) - \frac{1}{n_l} \sum_{r \in G_l} d^2(V_j, V_r)\right],$$

we have
$$L \geq d^2(V^{(l)}, V^{(k)}) - d(V^{(l)}, V^{(k)})K_1 - K_2,$$
w.p. at least $(1 - c/n^2)$, where
$$K_1 = C\sqrt{\log nt}\mathcal{V}^{1/2} + Ct^2(p + \log \bar{n})\mathcal{V}Tv^{1/2},$$
$$K_2 = Ct^2p^2 \log n\mathcal{V},$$
for some constant $C, c$.

*Proof.* First note that we can decompose the term into three terms:
$$\frac{1}{n_k}\sum_{r \in G_k} d^2(V_j, V_r) - \frac{1}{n_l}\sum_{r \in G_l} d^2(V_j, V_r) = U_1 - U_2 + U_3,$$
where
$$U_1 := \frac{1}{n_k}\sum_{r \in G_k} d^2(V_j, V_r) - d^2(V_j, V^{(k)}),$$
$$U_2 := \frac{1}{n_l}\sum_{r \in G_l} d^2(V_j, V_r) - d^2(V_j, V^{(l)})$$
$$U_3 := d^2(V_j, V^{(k)}) - d^2(V_j, V^{(l)}).$$

If we further define $U_0 := \frac{1}{2}\left[\frac{1}{n_l^2}\sum_{s,r \in G_l} d^2(V_s, V_r) - \frac{1}{n_k^2}\sum_{s,r \in G_k} d^2(V_s, V_r)\right]$, then we have
$$L = U_0 + U_1 - U_2 + U_3.$$
From Corollary 12 we know $U_1$ and $U_2$ can be lower bounded by throwing out the remainders $\Delta_1, \Delta_2$, i.e.,

$$U_1 = \frac{1}{n_k}\sum_{r \in G_k} d^2(V_j, V_r) - d^2(V_j, V^{(k)})$$
$$\geq \left\langle \mathcal{A}(V_j, V^{(k)}), \frac{1}{n_k}\sum_{r \in G_k} t(X_r V^{(k)} + V^{(k)}X_r) + t^2 X_r V^{(k)} X_r \right\rangle,$$

$$U_3 = d^2(V_j, V^{(k)}) - d^2(V_j, V^{(l)})$$
$$\geq d^2(V^{(l)}, V^{(k)}) + \left\langle \mathcal{A}(V^{(l)}, V^{(k)}), t(X_j V^{(l)} + V^{(l)}X_j) + t^2 X_j V^{(l)} X_j \right\rangle$$
$$- d^2(V_j, V^{(l)}).$$

As for the $U_0$ and $U_3$, we choose to use triangle inequality to get a rough bound, i.e., by noting $d(V_j, V_r) \leq d(V_j, V^{(l)}) + d(V_r, V^{(l)})$, we have
$$U_2 = \frac{1}{n_l}\sum_{r \in G_l} d^2(V_j, V_r) - d^2(V_j, V^{(l)})$$
$$\leq \frac{1}{n_l}\sum_{r \in G_l} d^2(V^{(l)}, V_r) + \frac{2}{n_l}d(V^{(l)}, V_r)\sum_{r \in G_l} d(V_j, V^{(l)}).$$

And
$$U_0 = \frac{1}{2}\left[\frac{1}{n_l^2}\sum_{s,r \in G_l} d^2(V_s, V_r) - \frac{1}{n_k^2}\sum_{s,r \in G_k} d^2(V_s, V_r)\right]$$
$$\geq -\frac{1}{2}\frac{1}{n_l^2}\sum_{s,r \in G_k} (d(V_s, V^{(k)}) + d(V_r, V^{(k)}))^2$$
$$\geq -\frac{2}{n_l}\sum_{r \in G_k} d^2(V^{(k)}, V_r).$$

For the RHS of the inequality for $U_1$, it can be divided into two parts.

$$Z_1^1 := \left\langle \mathcal{A}(V_j, V^{(k)}), \frac{1}{n_k} \sum_{r \in G_k} t(X_r V^{(k)} + V^{(k)} X_r) \right\rangle$$

and

$$Z_2^1 := \left\langle \mathcal{A}(V_j, V^{(k)}), t^2 \frac{1}{n_k} \sum_{r \in G_k} X_r V^{(k)} X_r \right\rangle.$$

the first part is a Gaussian distribution whose variance can be bounded by $c_1 t^2 \|\mathcal{A}(V_j, V^{(k)}) V^{(k)}\|_F^2 / n_k$, for some constant $c_1$. By Gaussian tail bound $P(|N(0,1)| > u) \leq e^{-u^2/2}, \forall u > 0$ and Lemma 13, we have

$$|Z_1^1| \leq c_2 t \sqrt{\log n} \|V^{(k)}\|^{1/2} / \sqrt{n_k} \cdot d(V_j, V^{(k)})$$
$$\leq c_2 t \sqrt{\log n} \mathcal{V}^{1/2} \cdot d(V_j, V^{(k)}),$$

w.p. $\geq 1 - c_3/n^2$, for some constant $c_2, c_3$. On the other hand,

$$|Z_2^1| = t^2 \left| \left\langle \mathcal{A}(V_j, V^{(k)})(V^{(k)})^{1/2}, \frac{1}{n_k} \sum_{r \in G_k} X_r V^{(k)} X_r (V^{(k)})^{-1/2} \right\rangle \right|$$

$$\leq t^2 \left\| \frac{1}{n_k} \sum_{r \in G_k} X_r V^{(k)} X_r (V^{(k)})^{-1/2} \right\|_F \cdot d(V_j, V^{(k)})$$

$$\leq t^2 \frac{1}{n_k} \sum_{r \in G_k} \|X_r\|^2 \|V^{(k)}\| \left\| (V^{(k)})^{-1/2} \right\|_F \cdot d(V_j, V^{(k)})$$

$$\leq t^2 \max_{r \in G_k} \|X_r\|^2 \mathcal{V} \mathcal{T} v^{1/2} \cdot d(V_j, V^{(k)})$$

$$\leq c_4 t^2 (p + \log n) \mathcal{V} \mathcal{T} v^{1/2} \cdot d(V_j, V^{(k)}),$$

w.p. $\geq 1 - c_5/n^2$, for some constant $c_4, c_5$. The last inequality can be implied from union bound and Corollary 4.4.8 in Vershynin [2018]:

$$\|X_r\| \leq C(\sqrt{p} + u), \quad \text{w.p.} \geq 1 - 4e^{-u^2}.$$

Now by combining $Z_1^1, Z_2^1$ we have

$$U_1 \geq Z_1^1 + Z_2^1$$
$$\geq -\left[ c_2 t \sqrt{\log n} \mathcal{V}^{1/2} + c_4 t^2 (p + \log n) \mathcal{V} \mathcal{T} v^{1/2} \right] \cdot d(V_j, V^{(k)}),$$

w.p. $\geq 1 - (c_3 + c_5)/n^2$.

For $U_0$, we have

$$U_0 \geq -\frac{2}{n_k} \sum_{r \in G_k} d^2(V^{(k)}, V_r)$$

$$= -\frac{2t^2}{n_k} \sum_{r \in G_k} \text{Tr}(X_r V^{(k)} X_r)$$

$$\geq -2t^2 \mathcal{V} \frac{1}{n_k} \sum_{r \in G_k} \text{Tr}(X_r^2)$$

$$\geq -c_6 t^2 \mathcal{V} p^2,$$

w.p. $\geq 1 - c_7/n^2$ for some constant $c_6, c_7$. The equation is a direct result by definition of Wasserstein distance for Gaussians:

$$d^2(V^{(k)}, V_r) = \text{Tr}(V^{(k)}) + \text{Tr}(V_r) - 2\text{Tr}(\sqrt{(V^{(k)})^{1/2} V_r (V^{(k)})^{1/2}}).$$

Note here

$$\sqrt{(V^{(k)})^{1/2}V_r(V^{(k)})^{1/2}} = \sqrt{(V^{(k)})^{1/2}(I+tX_r)V^{(k)}(I+tX_r)(V^{(k)})^{1/2}}$$
$$= (V^{(k)})^{1/2}(I+tX_r)(V^{(k)})^{1/2}.$$

The last inequality can be derived through Bernstein's inequality (Theorem 2.8.2 in Vershynin [2018]) by noting that $\text{Tr}(X_r^2)$ is sub-exponential with mean $\mathbb{E}(\text{Tr}(X_r^2)) = p^2$. Similar to the argument for $U_0, U_1$, after we apply high-dimensional bound for sub-Gaussian or sub-exponential distributions we can get bound for $U_2, U_3$:

$$U_2 \le \frac{1}{n_l}\sum_{r\in G_l} d^2(V^{(l)}, V_r) + \frac{2}{n_l}\sum_{r\in G_l} d(V_r, V^{(l)})d(V^{(l)}, V_j)$$
$$\le c_8 t^2 \mathcal{V}p^2 \log n,$$

w.p. $\ge 1 - c_9/n^2$, for some constant $c_8, c_9$.

$$U_3 \ge d^2(V^{(l)}, V^{(k)}) + \left\langle \mathcal{A}(V^{(l)}, V^{(k)}), t(X_j V^{(l)} + V^{(l)} X_j) + t^2 X_j V^{(l)} X_j \right\rangle$$
$$- d^2(V_j, V^{(l)})$$
$$\ge d^2(V^{(l)}, V^{(k)}) - \left[ c_2 t\sqrt{\log n}\mathcal{V}^{1/2} + c_4 t^2(p+\log\bar{n})\mathcal{V}Tv^{1/2} \right] \cdot d(V^{(l)}, V^{(k)})$$
$$- c_{10} t^2(p+\log n)p\mathcal{V},$$

w.p. $\ge 1 - c_{11}/n^2$., for some constant $c_{10}, c_{11}$. Lastly, by noting $d(V_j, V^{(k)}) \le d(V^{(l)}, V^{(k)}) + d(V_j, V^{(l)})$ in $U_1$, and combine them together we have

$$L = U_0 + U_1 - U_2 + U_3$$
$$\ge d^2(V^{(l)}, V^{(k)}) - d(V^{(l)}, V^{(k)})K_1 - K_2,$$

w.p. at least $(1 - c/n^2)$, where

$$K_1 = C\sqrt{\log n}t\mathcal{V}^{1/2} + Ct^2(p+\log\bar{n})\mathcal{V}Tv^{1/2},$$
$$K_2 = Ct^2 p^2 \log n\mathcal{V},$$

for some constant $C, c$. ∎

*Lemma* 15 ($\tilde{E}_{1,j}^{(k,l)}$ **upper bound**). Suppose $v \in \Gamma_K := \text{span}\{1_{G_k} : k \in [K]\}^{\perp}, \|v\| = 1$. Let

$$E_{1,j}^{(k,l)} = \frac{1}{n_k}\sum_{r\in G_k} d^2(V^{(k)}, V_r) + \left[ \frac{2}{n_k}\sum_{r\in G_k} d(V^{(k)}, V_r)d(V_j, V^{(k)}) \right],$$

and $\tilde{E}_{1,j}^{(k,l)} = E_{1,j}^{(k,l)}/d(V^{(l)}, V^{(k)})$. Then w.p. $\ge 1 - n^{-2}$, we have

$$\sum_{j\in G_l} \tilde{E}_{1,j}^{(k,l)}|v_j| \le C\mathcal{V}^{1/2} pt\sqrt{n_l}\left(\sum_{j\in G_l} v_j^2\right)^{1/2},$$

*Proof.* Note $E(\text{Tr}(X_r^2)) = p^2, E(\sqrt{\text{Tr}(X_r^2)}) \le \sqrt{E(\text{Tr}(X_r^2))} = p$ by Jensen's inequality. From high-dimension bound for sub-exponential and sub-Gaussian (Hoeffding's inequality and Bernstein's inequality) we have that w.p. $\ge 1 - c/n^2$,

$$\frac{1}{n_k}\sum_{r\in G_k} d^2(V^{(k)}, V_r) = \frac{1}{n_k}\sum_{r\in G_k} \text{Tr}(X_r^2 V^{(k)}) \le C\mathcal{V}p^2 t^2,$$
$$\frac{1}{n_k}\sum_{r\in G_k} d(V^{(k)}, V_r) = \frac{1}{n_k}\sum_{r\in G_k} \sqrt{\text{Tr}(X_r^2 V^{(k)})} \le C\mathcal{V}^{1/2} pt,$$

for some constants $C, c$. Suppose that $d(V^{(l)}, V^{(k)}) \ge C_0 t\mathcal{V}^{1/2}\sqrt{\log n}p$, for some fixed constant $C_0$. Then we have w.p. $\ge 1 - c/n^2$

$$d(V_j, V^{(k)}) \le d(V_j, V^{(l)}) + d(V^{(l)}, V^{(k)}) \le Ctp\sqrt{\log n}\mathcal{V}^{1/2} + d(V^{(l)}, V^{(k)}) \le Cd(V^{(l)}, V^{(k)}),$$

for some large constant $C$. So we have w.p. $\geq 1 - c/n^2$

$$\sum_{j \in G_l} \tilde{E}_{1,j}^{(k,l)} |v_j| \leq C\mathcal{V}^{1/2} pt \sum_{j \in G_l} |v_j| \leq C\mathcal{V}^{1/2} pt \sqrt{n_l} \left( \sum_{j \in G_l} v_j^2 \right)^{1/2},$$

where $\mathcal{V} = \max_k \|V^{(k)}\|$, for some large constant $C$. $\blacksquare$

*Lemma* 16 ($\tilde{E}_{2,j}^{(k,l)}$ **upper bound**). Suppose $v \in \Gamma_K := \text{span}\{1_{G_k} : k \in [K]\}^\perp, \|v\| = 1$. Let

$$E_{2,j}^{(k,l)} = d^2(V^{(l)}, V_j) + 2d(V^{(l)}, V_j)d(V^{(l)}, V^{(k)}).$$

and $\tilde{E}_{2,j}^{(k,l)} = E_{2,j}^{(k,l)}/d(V^{(l)}, V^{(k)})$. Then w.p. $\geq 1 - n^{-1}$, we have

$$\sum_{j \in G_l} \tilde{E}_{2,j}^{(k,l)} |v_j| \leq Ct\mathcal{V}^{1/2} p(\sqrt{n_l} + \log^2(n)) \left( \sum_{j \in G_l} v_j^2 \right)^{1/2},$$

*Proof.* First we make the following claim:

*Claim* 17. Following the above setting, w.p. $\geq 1 - cn^{-1}$, we have

$$\sum_{j \in G_l} d^2(V^{(l)}, V_j)|v_j| \leq Ct^2 \mathcal{V} p^2 (\sqrt{n_l} + \log(n)^2) \left( \sum_{j \in G_l} v_j^2 \right)^{1/2}, \tag{24}$$

$$\sum_{j \in G_l} d(V^{(l)}, V_j)|v_j| \leq Ct\mathcal{V}^{1/2} p \sqrt{n_l} \left( \sum_{j \in G_l} v_j^2 \right)^{1/2}, \tag{25}$$

for some large constant $C$.

If the claim holds, by plugging in the lower bound for $\Delta$ in the assumption, we have

$$\sum_{j \in G_l} \tilde{E}_{2,j}^{(k,l)} |v_j| \leq \frac{Ct^2 \mathcal{V} p^2 (\sqrt{n_l} + \log(n)^2)}{C_0 t \mathcal{V}^{1/2} p \sqrt{\log n}} \left( \sum_{j \in G_l} v_j^2 \right)^{1/2} + Ct\mathcal{V}^{1/2} p \sqrt{n_l} \left( \sum_{j \in G_l} v_j^2 \right)^{1/2}$$

$$\leq Ct\mathcal{V}^{1/2} p(\sqrt{n_l} + \log^2(n)) \left( \sum_{j \in G_l} v_j^2 \right)^{1/2}$$

*Proof of the claim.* First we look at (25):

$$\sum_{j \in G_l} d(V^{(l)}, V_j)|v_j| \leq t\mathcal{V}^{1/2} \sum_{j \in G_l} \sqrt{\text{Tr}(X_j^2)} |v_j|.$$

By Theorem 2.6.3 (General Hoeffding's inequality) in Vershynin [2018] we have w.p.$\geq 1 - c/n^2$,

$$\sum_{j \in G_l} \sqrt{\text{Tr}(X_j^2)} |v_j| \leq p \sum_{j \in G_l} |v_j| + Cp\sqrt{n_l} \left( \sum_{j \in G_l} v_j^2 \right)^{1/2},$$

for some constant $C$. i.e., w.p.$\geq 1 - c/n^2$,

$$\sum_{j \in G_l} d(V^{(l)}, V_j)|v_j| \leq Ct\mathcal{V}^{1/2} p \sqrt{n_l} \left( \sum_{j \in G_l} v_j^2 \right)^{1/2},$$

for some constant $C$. Next we will show (24). First note that $d^2(V^{(l)}, V_j) \le t^2 \mathcal{V} \mathrm{Tr}(X_j^2)$, let

$$G_1(v) = \left| \sum_{j \in G_l} [\mathrm{Tr}(X_j^2) - \mathbb{E}\mathrm{Tr}(X_j^2)]|v_j| \right|,$$

then

$$\sum_{j \in G_l} d^2(V^{(l)}, V_j)|v_j| \le t^2 \mathcal{V} G_1(v) + t^2 \mathcal{V} p^2 \sqrt{n_l} \left( \sum_{j \in G_l} v_j^2 \right)^{1/2}.$$

W.O.L.G., we may assume $v \in \mathbb{V} := \{v \in \Gamma_K : \|v\| = 1\}$, $\|G_1\|_{\mathbb{V}} := \sup_{v \in \mathbb{V}} |G_1(v)|$. Then by Theorem 4 in Adamczak [2008] we know

$$\mathbb{P}(\|G_1\|_{\mathbb{V}} \ge 2\mathbb{E}\|G_1\|_{\mathbb{V}} + s) \le \exp\left(-\frac{s^2}{3\tau_1^2}\right) + 3\exp\left(-\frac{s}{3\|M_1\|_{\psi_1}}\right),$$

where

$$\tau_1^2 = \sup_{v \in \mathbb{V}} \sum_{j \in G_l} v_j^2 \mathbb{E}[\mathrm{Tr}(X_j^2) - \mathbb{E}\mathrm{Tr}(X_j^2)]^2 \le \mathbb{E}[\mathrm{Tr}(X_j^2)]^2 \le p^4,$$

$$M_1 = \max_{j \in G_l, v \in \mathbb{V}} \left| v_j[\mathrm{Tr}(X_j^2) - \mathbb{E}\mathrm{Tr}(X_j^2)] \right| \le \max_{j \in G_l} \left| [\mathrm{Tr}(X_j^2) - \mathbb{E}\mathrm{Tr}(X_j^2)] \right|.$$

By maximal inequality (Lemma 2.2.2 in van der Vaart and Wellner [1996]) we have

$$\|M_1\|_{\psi_1} \le C \log(n_l) \max_{j \in G_l} \left\| [\mathrm{Tr}(X_j^2) - \mathbb{E}\mathrm{Tr}(X_j^2)] \right\|_{\psi_1} \le C \log(n_l) p^2.$$

So by choosing $s = C \log^2(n)p^2$, we have w.p. $\ge 1 - c/n$,

$$G_1(v) \le 2\mathbb{E}\|G_1\|_{\mathbb{V}} + C \log^2(n)p^2,$$

for some $C, c$. On the hand,

$$\begin{aligned}
\mathbb{E}\|G_1\|_{\mathbb{V}} = \mathbb{E} \left| \sum_{j \in G_l} [\mathrm{Tr}(X_j^2) - \mathbb{E}\mathrm{Tr}(X_j^2)]|v_j| \right| \\
\le \sum_{j \in G_l} \mathbb{E}|\mathrm{Tr}(X_j^2) - \mathbb{E}\mathrm{Tr}(X_j^2)||v_j| \\
\le 2\mathbb{E}|\mathrm{Tr}(X_1^2)|\sqrt{n_l} \left( \sum_{j \in G_l} v_j^2 \right)^{1/2} \\
= 2p^2 \sqrt{n_l} \left( \sum_{j \in G_l} v_j^2 \right)^{1/2}.
\end{aligned}$$

So w.p. $\ge 1 - cn^{-1}$, we have

$$\sum_{j \in G_l} d^2(V^{(l)}, V_j)|v_j| \le Ct^2 \mathcal{V} p^2 (\sqrt{n_l} + \log(n)^2) \left( \sum_{j \in G_l} v_j^2 \right)^{1/2}.$$

$\blacksquare$