# OpenReview forum: "Wasserstein $K$-means for clustering probability distributions"
_NeurIPS.cc/2022/Conference — NeurIPS 2022 Accept_

### Official Review · Reviewer_Arph · 2022-06-15

**Rating:** 5
**Confidence:** 3
**Soundness:** 3 good
**Presentation:** 3 good
**Contribution:** 3 good

**Summary:**

The paper performs studies on three different implementations of K-means clustering for probability distributions. Conventionally, the problem was solved by finding barycenters. The authors point out that such approaches are unsuitable because of two pitfalls in Wasserstein barycenters. In contrast, the alternative implementations with pairwise distances and SDP relaxation can provide better clustering results.


**Questions:**

The proposed distance-based and SDP-based approaches do not include barycenters or means as optimization variables. Can they still be called "K-means"?


Although the proposed alternatives work better in the given examples, they are more expensive for large-scale data sets because they have a quadratic cost to the number of instances. Optimizing over the cluster membership also makes them more difficult to parallelize. All given examples are small and cannot reveal the drawback. The whole MNIST data set has 70,000 images, but the authors picked only a few hundred.

How did the authors obtain the pairwise distances for MNIST?

**Limitations:**

In the checklist, the authors claimed that they described the limitations of their work but did not say where. There is no discussion or conclusion section in the paper, and I cannot find any relevant part for the limitations.

**Strengths And Weaknesses:**

Strength:
The work points out the difference between Wasserstein K-means and conventional K-means. Good examples are provided. Through the paper, readers can realize the difficulty due to the Wasserstein barycenters, and future research efforts should be devoted to non-barycenter-based clustering algorithms.

Weakness:
Most illustrative examples are small and artificial. Although distance-based and SDP-based approaches work better for such examples, it is hard to say they are the future for the research problem because they are not scalable.

---

> ### Author Response · Authors · 2022-08-02
> **We thank you for your positive comments on our work and appreciate all your precious advice.**
>
> (__On scalability of our approaches__) The scalability issue is one of our concerns. As we pointed out in Appendix B, we may consider several methods to bring down the time cost e.g. subsampling-based method for K-means and SDP. In the revision, we can observe the time complexity issue for Wasserstein $K$-means methods. Thus, we might consider more possible ways to accelerate the calculation for Wasserstein distance in the future.
>
> \
> (__On the `$K$-means' terminology__) We follow the convention in the SDP relaxed formulations of the Euclidean $K$-means clustering methods. In the Euclidean case, the distance-based (and its SDP formulation) and centroid-based formulations are equivalent, the SDP formulation is usually called _SDP relaxed $K$-means_. Even though the two formulations are not equivalent in Wasserstein space, we still abide this convention for the reason we have similar SDP structures.
>
> \
> (__On experiments for real datasets__) Please refer to our response item ``On experiments for real datasets" to reviewer k4tW.
>
> \
> (__On the pairwise distances for real dataset__) We use Sinkhorn divergence to calculate the pairwise distances for MNIST the same as Experiment 4.1. Sinkhorn distance approximates the Wasserstein distance fast by adding an entropy term to the original optimal transport. And Sinkhorn divergence is a debiased form of Sinkhorn distance. One may refer to the paper _Sinkhorn Distances: Lightspeed Computation of Optimal Transport_ by Cuturi, and the paper _Learning Generative Models with Sinkhorn Divergence_ by Genevay, Peyré, Cuturi for more details.
>
> \
> (__On adding discussion session__) We have now added a discussion section in the rebuttal revision, which discussed some limitations and future directions.

---

### Official Review · Reviewer_Ln53 · 2022-07-12

**Rating:** 7
**Confidence:** 4
**Soundness:** 3 good
**Presentation:** 3 good
**Contribution:** 3 good

**Summary:**

Paper studies the behavior of two formulations of Wasserstein-K-means: i) a centroid based ii) a distance based formulation. It shows that the centroid-based formulation can have some unexpected behaviors and illustrates the phenomenon on several toy examples. A SDP formulation of the distance based formulation is then proposed in order to speed up the computations. Experiments on simulated data are given, together with a simple real-data application on a MNIST clustering scenario.

**Questions:**

See above

**Limitations:**

As far as I can see, there is no potential negative societal impact of the work.

**Strengths And Weaknesses:**

Strengths of the paper :
- the paper highlight an interesting behavior of the Wasserstein-k-means, advocating for the use of the distance-based formulation rather than the centroid based one (exhibiting some cases in which they lead to different solutions in the Wasserstein space). To my knowledge, this analysis is new.
- Illustrations are provided, allowing better understanding the behavior in some specific cases.

Weaknesses:
- The paper advocates the use of the distance-based formulation but the scenarii that are considered seem rather limited. An in-depth analysis of the cases in which the difference appears and matters is lacking. For instance, it seems that it is related to « unbalanced » scenario, that is to say when the different cluster sizes are different. This behavior should be better investigated.
- the complexity analysis of the algorithm is not provided.
- The experimental section is rather limited, with one simple illustration on a MNIST clustering (digit 0 vs. digit 5) provided.

---

> ### Author Response · Authors · 2022-08-02
> **We thank you for your positive comments on our work and appreciate all your valuable comments.**
>
> (__On in-depth analysis of the superior behavior for our approaches__) Yes, it is a good point that data imbalance might be the primary reason for bad clustering accuracy. However, we observe that in our counterexample (Example 3) of illustrating the failure of centroid-based Wasserstein $K$-means, the failure is actually not due to the cluster imbalance. In fact, if we always set the number of copies of $\mu_4$ to be equal to $2m+1$, where $m$ is number of copies of $\mu_1$ and $\mu_2$, then by Lemma 4 we know centroid based $K$-means will fail given sufficiently large $m$ while those two clusters are balanced.
> \
> In our real data example, the superior behavior of distanced-based Wasserstein $K$-means and its SDP relaxation would be more evident for cases with unbalanced cluster sizes as you observed. However, the core reason should be the variability of probability measures within the true clusters. Intuitively speaking, the larger variability one cluster possesses, the more likely it would be for us to obverse the peculiar behaviors of barycenter-based Wasserstein $K$-means methods since the barycenters are no longer representing the clusters due to their instability or irregularity as we discussed in Section 2.1. In particular, in the MNIST dataset, we find that the within-cluster variability is partly from different orientations of the same digit, which leads to the worse performance of D-WKM.
>
> \
> (__On complexity analysis of the algorithm__) If you are referring to the theoretical analysis, please find the discussion at the beginning of the next response item ``On experiments for real datasets", which is now also discussed in Appendix B.
> If you are referring to the run time analysis for the numerical experiments, we have now added the time cost for the real datasets. For example, from Table 4 we can observe that all three Wasserstein $K$-means approaches have time complexity issues when we enlarge $n$. The nearly quadratic grow of time costs for W-SDP and D-WKM is due to the fact that calculating pairwise distances is the computational bottleneck for sample size of order $10^2$. The large variance for B-WKM for larger sample size from Table 3 is due to the convergence of the algorithm, where the total iterations for B-WKM achieves maximum iteration 100 for 2 replicates out of 10 total replicates.
>
> \
> (__On experiments for real datasets__) Please refer to our response item ``On experiments for real datasets" to reviewer k4tW.

---

### Official Review · Reviewer_dPcg · 2022-07-20

**Rating:** 6
**Confidence:** 3
**Soundness:** 3 good
**Presentation:** 2 fair
**Contribution:** 2 fair

**Summary:**

The paper uses k-means like algorithm to cluster probability measures using 2- Wasserstein metric. k-means clustering can be formulated in two ways 1) Centroid Based and 2) Distance based . Both formulations are equivalent in the Euclidean space however in the Wasserstein space the same is not true. The authors show that in the Wasserstein space centroid based k-means is much inferior to the distance based version due to irregularity and non-robustness. This is shown using well crafted examples. The authors empirically demonstrate the superiority of the distance based version. In addition the authors also give an SDP relaxation of the distance based k-means in Wasserstein space and show both theoretically and empirically that when the clusters are well separated , the SDP relaxation formulation can recover the clusters exactly with high probability.

**Questions:**

1)Did you compare the empirical results with Euclidean k-means algorithm as baseline, used on the probability distribution vectors?
2) Do you try some experiments with distributions other than Gaussian? Are there any results on exact recovery for other probability measures?

**Limitations:**

see the above sections

**Strengths And Weaknesses:**

Strengths:
1) The paper appears technically sound. I did not check the proofs but the main theorem and other results appear correct.
2) Clustering is a very important problem in Machine learning and hence extending a popular algorithm like k-means to another non-Euclidean metric space, backed with a nice theoretical result will be of interest to the community.
3) The use of examples with accompanied figures to highlight the important results is appreciable and aids understanding.

Weaknesses:
1) The paper is difficult to follow for a person not having background knowledge of Wasserstein space. For example terms like Alexandrov curvature, optimal transport map etc. are used without giving additional information or definition. I believe adding a bit more background information and details of notations used will go a long way in improving the readability of the paper for broader audience.
2) Experiments are done on very small data and also only for k values 2 and 4. More experiments with different values of k and larger datasets (may be more clusters from MNIST ) will strengthen the paper.
3) Related work for clustering of probability distributions is discussed very briefly. It would be good if the authors can cite works on clustering probability distributions in general (if available) and compare and contrast the Wasserstein k-means method.

---

> ### Author Response · Authors · 2022-08-02
> **We thank you for your positive comments on our work and appreciate all your useful comments.**
>
> (__On optimal transport background__)
> We added some background introduction about optimal transport in Appendix C in the rebuttal revision.
>
> \
> (__On real data experiments__)
> Please refer to our response item ``On experiments for real datasets" to reviewer k4tW.
>
> \
> (__On more discussions about clustering probability measures literature__) In fact, clustering probability measures is quite a new topic, there have not been many works related to it yet. We adopted your suggestion by adding more discussions on related work for clustering of probability distributions in the rebuttal revision. The concept of clustering general measure-valued data is introduced by Domazakis et al. [2019], where the authors proposed the centroid-based Wasserstein $K$-means algorithm. Verdinelli and Wasserman
> [2019] proposed a modified Wasserstein distance for distribution clustering. And after that, Chazal et al. proposed a method in Clustering of measures via mean measure quantization by first vectorizing the measures in a finite Euclidean space followed by an efficient clustering algorithm such as single-linkage clustering with $L_\infty$ distance. The vectorization-based methods could improve the computational efficiency but might not be able to handle important aspects, such as shapes and orientations, of probability measures compared to clustering algorithms based the Wasserstein metric. Thus, vectorization-based method might fail for cases where problems are ill-conditioned, e.g., cluster sizes are highly unbalanced, or the within-cluster variability is high.
>
> \
> (__On comparing with Euclidean $K$-means__)
> We have added some numerical results based on the Euclidean $K$-means algorithm in the revision for real datasets.
> From Table 2, Table 5 and Table 6 we can observe that the Euclidean $K$-means algorithm consistently performs the worst for most cases since the original $K$-means cannot properly handle shapes and orientations of probability measures. For example, small perturbations of shapes and orientations of input images may drastically increase their mutual vectorized-Euclidean distances while will only incur small changes in the Wasserstein distances.
>
> \
> (__On experiments and theories for non-Gaussian distributions__) We conducted the theoretical investigation of exact recovery for Gaussian distributions since the Wasserstein distance between Gaussians have closed form expressions, and yet they are rich enough to reveal non-trivial theoretical findings. To our knowledge, our results on Gaussians are the first about exact recovery of clustering distributions. We did simulations using Gaussians in order to numerically verify the theorem findings. All real data examples, including the MNIST and other datasets we added in the rebuttal revisions, are not Gaussians, although cluster labels in these examples cannot be exactly recovered.

---

### Official Review · Reviewer_k4tW · 2022-07-23

**Rating:** 4
**Confidence:** 2
**Soundness:** 2 fair
**Presentation:** 3 good
**Contribution:** 2 fair

**Summary:**

In this paper, the authors provided the pitfalls of Wasserstein K-means via giving the examples and theoretical proof, and proposed a distance-based formulation of K-means in the Wasserstein space.  In addition, the authors conduct a set of experiments on real and simulation data examples to evaluate the proposal. The main contributions are:
1. The authors analyze the irregularity and non-robustness of barycenter-based formulation of K-means theoretically and empirically.
2. To escape the pitfalls of the Wasserstein space, the authors generalize the distance-based formulation of K-means to the Wasserstein space.


**Questions:**

1. Please further add the experiments of this paper.
2. Please include the detailed proof of the pitfalls.
3. The conclusion section may be missing. Please include this section in the end of this paper.


**Limitations:**

In this paper, the authors only consider the two categories problem, and conduct the experiments on the real-world dataset. The multiple categories problem also should be considered.

**Strengths And Weaknesses:**

Strengths:
1. The proposed problem is worth exploring
2. The research is meaningful
Weaknesses:
1. In this paper, the authors claim that they provide evidence for pitfalls of barycenter-based Wasserstein K-means, but I cannot find the details of proof for the irregularity and non-robustness.
2. Experiments are not enough. My main concern is the experiments in this paper. The authors conduct a set of experiments on real and simulation data examples to evaluate the proposal. However, the authors only conduct experiment on one real-world dataset and use error rate to evaluate the proposal. Using more datasets, and considering the imbalanced datasets, adding the F1 score can be more convincing.

---

> ### Author Response · Authors · 2022-08-02
> **We thank you for your comments on our work and appreciate all your helpful feedback.**
>
> (__On irregularity and non-robustness proofs__) Details about the non-robustness of Wasserstein barycenters are provided in Section 2 Example 2, where all calculations are straightforward as described therein. This example illustrates that small perturbation on the input distributions may incur dramatic changes in their barycenter. Our Example 1 illustrating the irregularity of Wasserstein barycenters is from Santambrogio and Wang [2016], where the barycenter of two convex-supported distributions is no longer convex-supported and therefore not preserving the shape; for self-contained purpose, we have added a derivation in Appendix D.1 in the rebuttal revision for the form of barycenter in Example 1.
>
> \
> (__On experiments for real datasets__) The scalability would be a major issue here and we are only able to focus on the experiments with sample size of the order of $10^2$ to $10^3$ in the current work.  As we pointed out in Appendix B, we may consider several methods to bring down the time cost e.g. subsampling-based method for $K$-means and SDP, which however is beyond the scope of the current work. We might consider more possible ways to accelerate the calculation for Wasserstein distance in the future.
> More specifically, the time cost of the standard and most state-of-the-art Sinkhorn algorithms for calculating optimal transport is of the order $O(g^4/\epsilon^2)$, and for calculating Wasserstein barycenters is $O(ng^4/\epsilon^2)$, where $n$ is the sample size,  $\epsilon$ is the numerical accuracy, and $g^2$ is the discretization size for a $g$-by-$g$ image, e.g. $g=28$ for MNIST datasets. Thus, it costs time of order $O(10^{16})$ to calculate the Wasserstein distance between two pictures with scale $100\times 100$ to achieve $10^{-4}$ accuracy. In practice, it takes 3.995 seconds to calculate the Wasserstein distance between two pictures with scale $36\times 36$ using Sinkhorn divergence in our settings, which indicates that we need to wait at least 11 hours to get a single result from the barycenter-based Wasserstein $K$-means (B-WKM) with $10$ total iterations and sample size $n=1000$ apart from the time cost for calculating barycenters. Moreover, it takes at least 45 days to get one result by distance-based $K$-means approach (D-WKM) or its semidefinite program relaxation (W-SDP) from the same settings.
> \
> As for the cluster numbers, our simple experiment with two clusters already shows the inferiority of the barycenter based B-WKM algorithm, which we feel already provide strong numerical evidence to backup our theoretical arguments; the inferior behaviors for B-WKM compared to D-WKM and W-SDP are expected to be more evident as the problem becomes more complicated. However, we need the sample size $n$ to be of the order at least $10^3$ to conduct convincing experiments with more clusters, which may not be accessible for now. On the other hand, the goal of our article is not trying to provide a more efficient way to cluster probability measures but to observe and analyze the peculiar behaviors of Wasserstein barycenters and propose the D-WKW and its semidefinite program relaxation W-SDP.
> \
> Nevertheless, we did some preliminary works to hopefully address your concerns partially. We have now included more experiments with two new real-world datasets, Fashion-MNIST and USPS handwriting digits, in the rebuttal revision. Both new experiments involve three clusters with unbalanced cluster sizes. We can observe similar patterns on the performance of different methods (Table 5 and Table 6) as that of the MNIST dataset. As we can see, the mis-classification error for the two distance based methods, W-SDP and D-WKM, are consistently and strictly better than the B-WKM by a fairly large margin. From Table 2 in the revision, we can observe that the results with doubled sample size are quite close to the results from the original experiment. Therefore, we can expect behaviors for Wasserstein $K$-means approaches to be stable and consistent as sample size grows.
> \
> Finally, we shall emphasize that the reason why we use error rate or mis-classification error as our clustering criterion is its convenience to interpret. The mis-classification error is the proportion of the data that are assigned to the wrong clusters; so there is no need to consider the imbalance of the clusters as the way for $F_1$ score with more than $2$ clusters. Nevertheless, we shall adopt your suggestions and add the average weighted $F_1$ score later. Since the time for rebuttal session is limited, we give the results for case 1 in Table 3 in the revision, where we can see consistent patterns between $F_1$ scores and error rates from Table 2. We will add the $F_1$ score as another criterion for the rest real datasets experiments in the future version.
>
> \
> (__On adding conclusion section__) We have now added a discussion section including the conclusion in the revision.

---

### Meta-Review · Area_Chair_aLEZ · 2022-08-23

**Recommendation:** Accept
**Confidence:** Less certain

**Metareview:**

This paper provides a Wasserstein-based k-means formulation for clustering probability distributions. Though the overall reception was mildly positive but two reviewers raised their scores following the author feedback. There remains some doubts that there is enough evidence to support all claims in the paper--most relevant, referees have noted to us that fundamentally a strong empirical validation is lacking. We recommend revising the paper and focusing on a strong empirical narrative (without relegating new details to the appendix), together with improving the exposition as outlined by reviewers

**Award:**

No

---

### Decision · Program_Chairs · 2022-09-14

Accept